materials science/biomaterials

polyurethane adhesive, dental restoration, hydrolysis-resistant, stress-buffering

**Authors for correspondence:**
Song Zhu
e-mail: zhusong1965@163.com
Zhanchen Cui
e-mail: cuizc@jlu.edu.cn

This article has been edited by the Royal Society of Chemistry, including the commissioning, peer review process and editorial aspects up to the point of acceptance.

# Hydrolysis-resistant and stress-buffering bifunctional polyurethane adhesive for durable dental composite restoration

Jiahui Zhang[1], Xiaowei Guo[1], Xiaomeng Zhang[1], Huimin Wang[1], Jiufu Zhu[2], Zuosen Shi[2], Song Zhu[1] and Zhanchen Cui[2]

[1]Department of Prosthetic Dentistry, Hospital of Stomatology, and [2]State Key Lab of Supramolecular Structure and Materials, College of Chemistry, Jilin University, Changchun 130012, People's Republic of China

JZhang, 0000-0002-2391-9381; SZ, 0000-0001-6935-9490; ZC, 0000-0002-3412-1594

A new elastic polyurethane (PU) adhesive was reported in this study to improve the stability and durability of the dental adhesion interface. A polyurethane oligomer was synthesized by the solution polymerization method, and a diluent and solvent were added to prepare PU adhesives. The water sorption, water solubility, contact angle, thermal stability, degree of conversion and mechanical properties of the PU adhesives were evaluated. Experimental applications for tooth restoration (microtensile bond strength and microleakage) were also performed, and cytotoxicity test was carried out. The water sorption and solubility of the PU adhesives were significantly lower than those of three commercial adhesives. The microtensile bond strength of the PU adhesives was improved after thermocycling test, and the extent of microleakage was diminished when compared with that of commercial adhesives. Biocompatibility testing demonstrated that the PU adhesive was non-toxic to L929 fibroblasts. This study shows the ability of PU adhesive to improve the stability and durability of the dental adhesion interface and may refocus the attention of scientists from rigid bonding to flexible bonding for dental adhesion, and it sheds light on a new strategy for the stable and durable bonding interface of dentine adhesives.

# 1. Introduction

Composite resin has been widely used in dental restoration for more than 60 years due to its aesthetic advantages, excellent mechanical properties, ease of use and acceptable price [1–4]. The success of composite resin restoration relies on bonding techniques that can bond these plastic materials to the tissue of teeth. Therefore, strong and durable bonding properties are necessary for successful composite resin restoration [5–8].

The failure of restorations is mainly due to defects in the bonding interface, which are caused by the polymerization stress when the composite resin is polymerized using a curing light [9]. Scientists have made many efforts to reduce the polymerization shrinkage of composite resins, for example, by using low-shrinkage composite resin. It has been reported that the volume of polymerization shrinkage of the composite resin can be reduced to less than 1%, and the generation of gaps between the tooth and the composite resin can be temporarily avoided [10,11]. However, there will be continuous mechanical chewing stress in the mouth after dental restoration. Moreover, studies have shown that the thermal expansion coefficient of the composite resin is usually $2.0 \times 10^{-3}\%\ ^\circ C^{-1}$, which is larger than that of dentine (approximately $1.1 \times 10^{-3}\%\ ^\circ C^{-1}$) [12]. During temperature changes, different volume expansion and contraction will occur repetitively between dentine and composite resins, which may cause stress within the adhesive layer and eventually lead to the occurrence of gaps in the bonding interface [13]. Bacteria and their acidic by-products, bacterial enzymes, liquids in the mouth and nutrients can penetrate into the interfacial gap, causing microleakage and eventually leading to demineralization of the teeth and secondary caries [14]. Therefore, we need to explore a new method to solve the problem of microleakage.

Since Kanca [15] introduced wet bonding technology into the field of dentine adhesive to prevent the collapse of demineralized collagen fibres of dentine, manufacturers have increased the concentration of hydrophilic monomers. For example, hydroxyethyl methacrylate (HEMA) can act as a solvent for mixing hydrophobic monomers to avoid phase separation and help the adhesive monomers to better penetrate into the dentinal tubule to form resin tags, forming a micromechanical interlocking [16–18]. However, hydrophilic resin monomers easily absorb water and are easily hydrolysed due to the presence of ester bond linkages. Therefore, increasing the content of the hydrophilic monomer in the adhesive will increase the water absorption of the polymer network, resulting in degradation of the adhesive layer, lowering the mechanical properties and ultimately leading to failure of the restoration [8,19–21].

Therefore, the study of adhesives with low water sorption and microleakage has attracted the attention of scientists striving to significantly improve the quality of the adhesive and reduce the failure of the restoration. Our research team has reported some methods. Cao *et al.* [22] prepared a superhydrophobic polyurethane coating to reduce microleakage. Yingchao and colleagues introduced a polyurethane elastic layer between the adhesive and the composite resin to buffer the stress generated by the restoration during use [13]. Gong *et al.* [23] synthesized a dual-curing polyurethane adhesive with carbon–carbon double bonds for conventional photocuring and the NCO group for continuously reacting with water molecules to improve bonding strength and durability. However, the NCO group is unstable in the air and is not easily stored, and the NCO group reacts with water to generate $CO_2$, which will generate bubbles in the adhesive layer, forming a weak point and jeopardizing the strength of the adhesive layer. Therefore, the adhesives synthesized in this study are all terminated with C=C for traditional photocuring. A polyurethane oligomer was synthesized by solution polymerization method using methylene-bis (4-cyclohexylisocyanate) and poly(tetrahydrofuran)1000/2000. The two types of synthetic polyurethane oligomers, the diluent and the solvent, were mixed in different proportions to prepare the adhesives, and the thermal stability, mechanical properties and biocompatibility were evaluated. It is desirable to obtain a polyurethane adhesive with a lower water absorption and solubility and elastic property that can buffer the stress within the adhesive, improving the stability and durability of the adhesive interface. A schematic model for this elastic PU adhesive used for tooth dentine bonding is illustrated in figure 1.

# 2. Experimental section

## 2.1. Materials

Spectrum Bond (Dentsply DeTrey GmbH, De-Trey-Strasse1, 78 467 Konstanz, Germany), Single Bond Universal (3 M ESPE, St Paul, MN, USA), Adper Single Bond 2 (3 M ESPE, St Paul, MN, USA) and Filtek Z350XT (3 M ESPE, St Paul, MN, USA) were used. Information on these three common

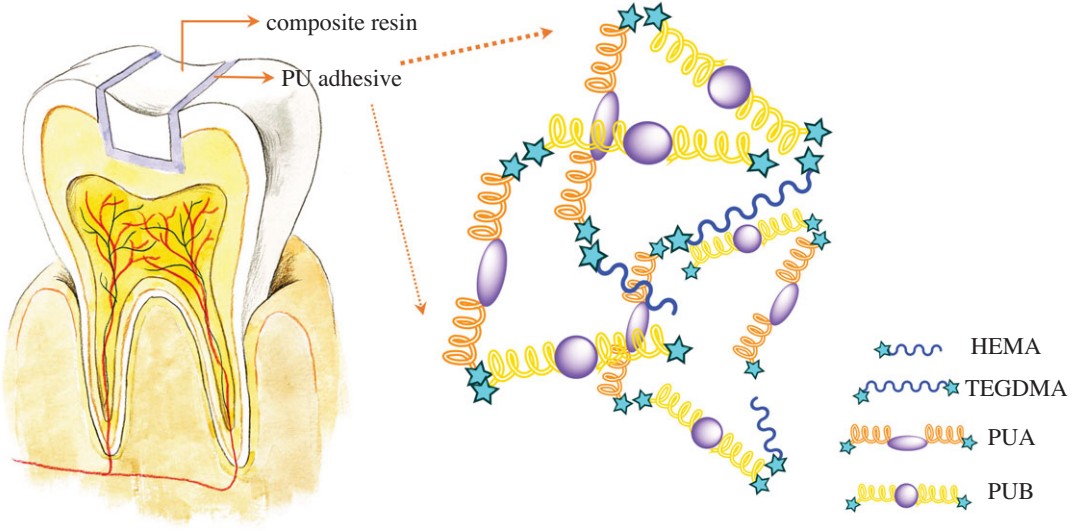

**Figure 1.** Schematic model for elastic PU adhesive using for tooth dentine bonding.

**Table 1.** Commercial adhesive for this study. bis-GMA: bisphenol a diglycidyl methacrylate, HEMA: 2-hydroxyethyl methacrylate, UDMA: urethane dimethacrylate, PENTA: phosphoric acid modified acrylate resin, BHT: butylhydroxytoluene, MDP: methacryloyloxydecyl dihydrogenphosphate. Lot number: SB2 (N912223); SPB (1801000919); SBU (4330297).

| material | code | category | formulation |
|---|---|---|---|
| single bond 2 | SB2 | 2-step etch-and-rinse | bis-GMA, HEMA, dimethacrylates, silica nanofiller, polyalkenoic acid copolymer, initiators, water, ethanol |
| spectrum bond | SPB | 2-step etch-and-rinse | UDMA, trimethacrylate, PENTA, highly dispersed silicon dioxide, camphorquinone, BHT, cetylamine hydrofluoride, acetone |
| single bond universal | SBU | universal adhesive | MDP phosphate monomer, bis-GMA, dimethacrylate resins, HEMA, Vitrebond copolymer, fillers, ethanol, water, initiators, silane |

commercial adhesives is shown in table 1. L929 cells were obtained from the School of Life Science, Jilin University. 3-(4,5-dimethylthiazole-2-yl)-2,5-diphenyltetrazolium bromide (MTT) was purchased from Sigma Aldrich. Methylene-bis(4-cyclohexylisocyanate) (HMDI), 2-hydroxyethyl methacrylate (HEMA), triethylene glycol dimethacrylate (TEGDMA), dibutyltin dilaurate (DBTDL), camphorquinone (CQ), ethyl-4-dimethylaminobenzoate (4-EDMAB) and methylene blue, acetone, tetrahydrofuran (THF), poly(tetrahydrofuran)1000/2000 (PTMEG1000/2000) were of analytical grade and were provided by Aladdin.

## 2.2. Preparation and characterization of polyurethane adhesive matrix

The synthesis process of polyurethane oligomer A (PUA) is shown in figure 2. First, HMDI (13.1175 g, 0.05 mol), DBTDL (0.09936 g, 3‰) and anhydrous THF were added into a three-necked round-bottom flask with a water-cooled condenser in a water bath of 70°C. PTMEG1000 (20 g, 0.02 mol) was then added into the reaction with mechanical stirring (500 r.p.m.) for 4 h with continuous $N_2$. When the hydroxyl groups disappeared, as monitored by infrared spectroscopy, HEMA (8.2 g) was added into the flask with continuous stirring for 3 h. The reaction was completed when the NCO group could not be detected by infrared spectroscopy. The liquid was precipitated with petroleum ether three times to give a white solid. The final products were dried in a vacuum oven to remove the remaining petroleum ether.

The synthesis process of polyurethane oligomer B (PUB) is shown in figure 2. First, HMDI (6.55875 g, 0.05 mol), DBTDL (0.07968 g, 3‰) and anhydrous tetrahydrofuran (THF) were added into a three-necked

**Figure 2.** Illustration of the fabrication process of the PU oligomer.

**Table 2.** Formulation of seven kinds of PU adhesives. PU adhesive = 70% PU (PUA + PUB) + 10% HEMA + 10% TEGDMA + 9% acetone + 0.3% CQ + 0.7% 4-EDMAB.

|      | PUA (g) | PUB (g) | HEMA (g) | TEGDMA (g) | Acetone (g) | CQ (g) | 4-EDMAB (g) |
|------|---------|---------|----------|------------|-------------|--------|-------------|
| PU1  | 12      | 0       | 1.71     | 1.71       | 1.54        | 0.05   | 0.12        |
| PU2  | 9       | 3       | 1.71     | 1.71       | 1.54        | 0.05   | 0.12        |
| PU3  | 8       | 4       | 1.71     | 1.71       | 1.54        | 0.05   | 0.12        |
| PU4  | 6       | 6       | 1.71     | 1.71       | 1.54        | 0.05   | 0.12        |
| PU5  | 4       | 8       | 1.71     | 1.71       | 1.54        | 0.05   | 0.12        |
| PU6  | 3       | 9       | 1.71     | 1.71       | 1.54        | 0.05   | 0.12        |
| PU7  | 0       | 12      | 1.71     | 1.71       | 1.54        | 0.05   | 0.12        |

round-bottom flask with a water-cooled condenser in a water bath of 70°C. PTMEG2000 (20 g, 0.01 mol) was then added into the reaction with mechanical stirring (500 r.p.m.) for 4 h with continuous N$_2$. When hydroxyl groups disappeared, as monitored by infrared spectroscopy, HEMA (4.2 g) was added into the flask with continuous stirring for 3 h. The reaction was completed until the NCO group could not be detected by infrared spectroscopy. The liquid was precipitated with petroleum ether three times to give a white solid. The final products were dried in a vacuum oven to remove the remaining petroleum ether.

Two types of polyurethane oligomers (PUA/PUB) are mixed as a matrix of adhesive at different mass ratios (12 : 0/9 : 3/8 : 4/6 : 6/4 : 8/3 : 9/0 : 12). The formulation is shown in table 2.

The structure of the polyurethane oligomer is characterized by Fourier transform infrared spectroscopy (FTIR) and nuclear magnetic resonance spectroscopy ($^1$H NMR spectrum). FTIR was measured by a Bruker Vertex 80 V infrared spectrometer in the range of 4000–500 cm$^{-1}$. The $^1$H NMR spectrum was measured by a Bruker Avance 500 MHz type III nuclear magnetic resonance spectrometer using deuterated chloroform as a solvent.

## 2.3. Water sorption and water solubility

Water absorption and water solubility were determined according to ISO 4049:2009. Disc-shaped samples ($d = 15.0$ mm, $h = 1.0$ mm, $n = 5$) were prepared. The polyurethane adhesive was poured into the mould, covered with a piece of polyester film and cured with a light intensity of 900 mW cm$^{-2}$ for 10 s. The curing light unit was monitored by a radiometer to ensure light intensity. All the samples prepared were placed in a desiccator with silica gel at $37 \pm 2$°C for 24 h. The samples were then transferred into another desiccator for 2 h at $23 \pm 1$°C and weighed. This process was repeated until a constant mass (M1) was obtained. The diameter and thickness of each sample were measured by

electronic digital caliper to calculate the volume (V; mm$^3$) of the sample. Each sample was then immersed in a sealed glass vial containing 15 ml of deionized water and soaked for 7 days at $37 \pm 1°C$. The samples were rinsed with running deionized water, and the surface water was dried with filter paper. Then, the samples were weighed to obtain mass M2. The samples were redried in a $37 \pm 1°C$ desiccator, as described above, until a stable mass M3 was obtained. The calculation formula for water absorption and solubility of the sample is as follows:

$$W_{SP} = \frac{M2 - M3}{V}$$

$$W_{SL} = \frac{M1 - M3}{V}$$

## 2.4. Contact angle measurements

Contact angles were obtained using the sessile drop method with a DataPhysics contact angle analyser (OCA-20, DataPhysics Co., Germany). This instrument consists of a CCD video camera with a resolution of $768 \times 576$ pixel and up to 50 images per second and multiple microsyringe units. A drop of 6 µl of deionized water was gently dropped onto the surface of the adhesive to take a digital photo. The digital drop image was processed by a specialized software SCA 20, which calculated both the left and right contact angles from the shape of the drop.

## 2.5. Tensile strength and elongation at break of polyurethane adhesives

The dumb-bell-shaped specimens ($n = 5$) were prepared in accordance with the standard ASTM-D638-2003. The prepared specimen is shown in figure 5$h$. The specimen was tested using a universal testing machine (AG-X plus, Shimadzu Corporation, Japan) with a cross-head speed of 10 mm min$^{-1}$ until it was broken.

## 2.6. Thermal stability characterization

Thermogravimetric analysis measurement of PU4 adhesive was performed using a TGA thermal analyser (Q500, TA Instruments, USA). Initial sample mass is around 5 mg. The heating rate is 10°C min$^{-1}$. The experiments were performed in an inert atmosphere with a continuous flow of nitrogen at the rate of 150 ml min$^{-1}$ and heated from room temperature to 800°C.

## 2.7. Degree of conversion

The degree of conversion (DC) of PU4 adhesives and three commercial adhesives were measured. The DC was determined by a Fourier transform infrared spectrometer equipped with an attenuated total reflectance device for five samples per group ($n = 5$). The FTIR of uncured adhesive was obtained as a control. The adhesive was cured for 10 s, and the polymerized adhesive was immediately subjected to FTIR. After light-curing, the area of infrared absorption peak of methacrylate double bonds (C=C, peak at 1637 cm$^{-1}$) decreased, and the carbonyl group (peak at 1720 cm$^{-1}$) was used as the internal standard. The calculation of the DC used the following equation:

$$DC\% = \left[ 1 - \frac{(A1636/A1720)\,\text{peak area after curing}}{(A1636/A1720)\,\text{peak area before curing}} \right] \times 100\%.$$

## 2.8. Microtensile bond strength test

The extracted teeth were stored in 1% chloramine T solution, placed at 4°C, and used within one month. The tooth was cut perpendicular to the long axis with slow-speed saw under water cooling in the middle section of the tooth to expose the dentine surface. Then, the dentine was sanded with 600 grit SiC paper to produce a uniform smear layer and was ultrasonically cleaned for 5 min. The prepared teeth were randomly divided into four groups (PU4 adhesive, SB2, SPB, and SBU groups). The specimen was etched with 37% phosphoric acid gel for 15 s, rinsed for 30 s and air-blown for 5 s. The adhesive was applied to the dentine surface using a microbrush, air-thinned for 5 s and light-cured for 10 s. Three 1.5 mm thick layers of commercial composite resin Z350XT were placed over the surface of the treated

dentine. Each resin composite was light-cured for 40 s using a light-curing unit. The specimens were soaked in deionized water at 37°C for 24 h. After immersion, the specimens were longitudinally cut into sticks of approximately 1.0 mm in width using a slow-speed saw. The dentine-resin stick was fixed to a microtensile mould using isocyanate glue. Then, it was tested on a universal testing machine with a cross-head speed of 1 mm min$^{-1}$. Microtensile strength was determined by dividing the loading force at break (N) by the cross-sectional area of the sticks (mm$^2$).

## 2.9. Scanning electron microscopy

The bonding surface of the PU4 adhesive and three commercial adhesives was detected by scanning electron microscopy (SEM; S4800, Hitachi Ltd, Tokyo, Japan). The specimen was sequentially grounded with 600 grit, 800 grit, 1200 grit, 2000 grit SiC paper under running water and ultrasonically cleaned for 10 min. Then, the specimen was etched with 37% phosphoric acid gel for 15 s and treated with 5.25% NaClO for 15 min followed by immersion in 50, 70, 90 and 100% ethanol for 15 min in sequence. Finally, the specimen was sprayed with platinum and observed by scanning electron microscopy.

## 2.10. Microleakage in composite restoration

Two standard class V cavities were prepared on the opposite surface of a molar (4 mm wide, 2.0 mm deep and 3.0 mm high) while a 45° edge bevel was prepared. The dentine was etched with 35% phosphoric acid gel for 15 s, rinsed for 15 s and continuously air-blown with condensed air. One side was randomly selected for the PU4 adhesive, and the other side received the commercial adhesive. The adhesive was directly applied for 20 s, lightly air-blown for 5 s and cured for 10 s. The cavity was filled with Filtek Z350XT composite resin layer by layer and cured for 40 s. The thickness of resin composite in four different groups was almost the same to ensure the reliability of the experiment. After polishing with sandpaper, the sample was stored in deionized water for 24 h. Artificial ageing was performed using a thermocycling instrument (PTC2c, Proto-tech, USA) for 5000 cycles between 5°C and 55°C baths with a dwell time of 30 s. After thermocycling was completed, the root apex was sealed with wax. The entire surface of the tooth was coated with transparent nail polish twice, except for the area within 1 mm of the tooth restoration interface. A microleakage test was conducted using a standard dye-leakage method. The prepared sample was placed in 1% methylene blue dye for 4 h at 37°C. The tooth surface was rinsed with deionized water and dried with filter paper. The crown portion was cut into a 1 mm sheet along the tooth long axis under running water cooling using a slow-speed diamond saw. The evaluation of microleakage was determined by evaluating the depth of dye into the tooth-restoration interface using a stereomicroscope. The depth of leakage of the dye was evaluated by the following criteria [22]:

0. no obvious dye leakage;
1. the dye gets to the interface to half the depth of the cavity;
2. leakage exceeds half of the depth of the hole but does not involve the axial surface;
3. leakage involves the axial surface but not the pulp; and
4. leakage involves the pulp.

## 2.11. Cytotoxicity test

The extracted solution was prepared by immersing the cured adhesive specimen in Dulbecco's modified Eagle's medium cell culture medium containing 10% fetal calf serum and 1% penicillin and streptomycin at a ratio of 3 cm$^2$ ml$^{-1}$ (the surface area of the specimen to the extracted solution volume) for 24, 48 and 72 h.

The L929 cells were cultured for an MTT assay. The cells were seeded in a 96-well plate at a density of $1.5 \times 10^4$ cells ml$^{-1}$ and incubated at 37°C in 5% CO$_2$ and 95% relative humidity for 24 h until the monolayer cells were spread over the bottom of the well. The original culture solution was replaced with 24, 48 and 72 h extracted solutions, and the control group was added to the cell culture medium with the surrounding wells sealed with phosphate buffered saline. Then, the cells were continuously incubated for 24 h and removed from the incubator. The morphology of the cells was observed under an inverted microscope. Mitochondrial dehydrogenase in living cells enabled the MTT to become insoluble formazan particles that can dissolve in dimethyl sulfoxide (DMSO). Next, 20 µl of MTT

solution (5 mg ml$^{-1}$) was added to each well, and the incubation was terminated after 4 h. Then, 150 µl of DMSO was added to each well, and the 96-well plate was shaken at a low speed for 10 min to fully dissolve the crystal formazan particles. The absorbance was read at a wavelength of 490 nm by a microplate reader (RT-6000, Lei Du Life Science and Technology Co., Shenzhen, China). The control group was regarded as the 100% cell proliferation rate, and the relative growth rate of each group was calculated.

The effect of PU4 adhesive on the activity of L929 cells was also evaluated using a live/dead cell staining kit. The cells were seeded at a density of $5 \times 10^4$ cells ml$^{-1}$ in a six-well plate for 24 h. After incubation with the 24 h extracted solution, the cells were stained with a live/dead cell staining kit according to the manufacturer's instructions. Live cells were stained green, and dead cells were stained red. The six-well plate was observed under a fluorescence microscope.

## 2.12. Statistical analysis

Data were expressed as the mean ± s.d. The data were consistent with normality and homoscedasticity distribution. Data for microtensile bond strength were analysed using two-way ANOVA, and the data of tensile strength, elongation at break, water solubility, water sorption, contact angle and degree of conversion were submitted to one-way ANOVA using SPSS software (v. 19.0, SPSS Inc., Chicago, IL, USA). Multiple comparison analysis was conducted using the Tukey test. The significance level was set at $p = 0.05$ for this study.

# 3. Results and discussion

## 3.1. Characterization of polyurethane oligomer

Figure 3a shows the $^1$H NMR spectrum of PUA. The peak at 7.29 ppm was attributed to the newly formed urethane group (-NHCOO-) after the reaction between HMDI and PTMEG. The protons belonging to the H of methylene (CH$_2$) from HEMA were clearly shown at 6.11 ppm. The resonance peak at 1.59 ppm was assigned to the signal of polytetrahydrofuran1000. The signal of methylene (CH$_2$) from HMDI appeared at 1.08 ppm. Figure 3b shows the $^1$H NMR spectrum of PUB, which was almost the same as that of PUA.

The PU oligomer (PUA/PUB) is synthesized by a conventional solution polymerization method. As shown in figure 4a of the infrared spectrum of the NCO-terminated PU prepolymer, the N-H stretching vibration at a wavenumber of 3340 cm$^{-1}$ and the C=O stretching vibration peak observed at 1725 cm$^{-1}$ were derived from the urethane group (-NHCOO-), which was attributed to the reaction of the NCO group with the OH group. The infrared stretching vibration peak of the NCO group can be seen at 2260 cm$^{-1}$. After the addition of HEMA, as shown in figure 4b, a C=C stretching vibration peak at 1636 cm$^{-1}$ was observed. The absorption peak of the NCO group at 2260 cm$^{-1}$ disappeared completely, indicating that all of the PU prepolymers reacted completely with HEMA. FTIR and $^1$H NMR spectra indicated that the PU oligomer (PUA/PUB) was successfully synthesized.

## 3.2. Water sorption and water solubility and contact angle measurements

The hydrophilic nature of a polymer depends largely on the chemical structure of the monomers and the linkage of the polymer. The most commonly used monomers in the dentine commercial adhesive system (HEMA, BPDM, MDP, bis-GMA) are hydrophilic monomers [24]. Moreover, an ester bond that is easily hydrolysed exists in the polymer formed by these hydrophilic monomers [24,25]. Therefore, it is more apt to absorb water. Since water molecules have a small molecular size and a high molar concentration, they can penetrate into the nanometre-size free volume space between polymer chains [26,27] or form clusters around functional groups (hydrophilic and ionic regions) to generate a hydrogen bond with the functional group [28]. These molecules are called bound water, which will break the hydrogen bond between the polymer chains, change the molecular structure and increase the mobility of partial polymer segment, leading to the swelling of the polymer [29], which plays a decisive role in the plasticization of the polymer [30,31]. Water absorption can lower the glass transition temperature, reduce thermal stability and deteriorate the mechanical properties of the polymer. It can be predicted that the strength of the adhesive connecting the dentine and the composite resin will decrease due to

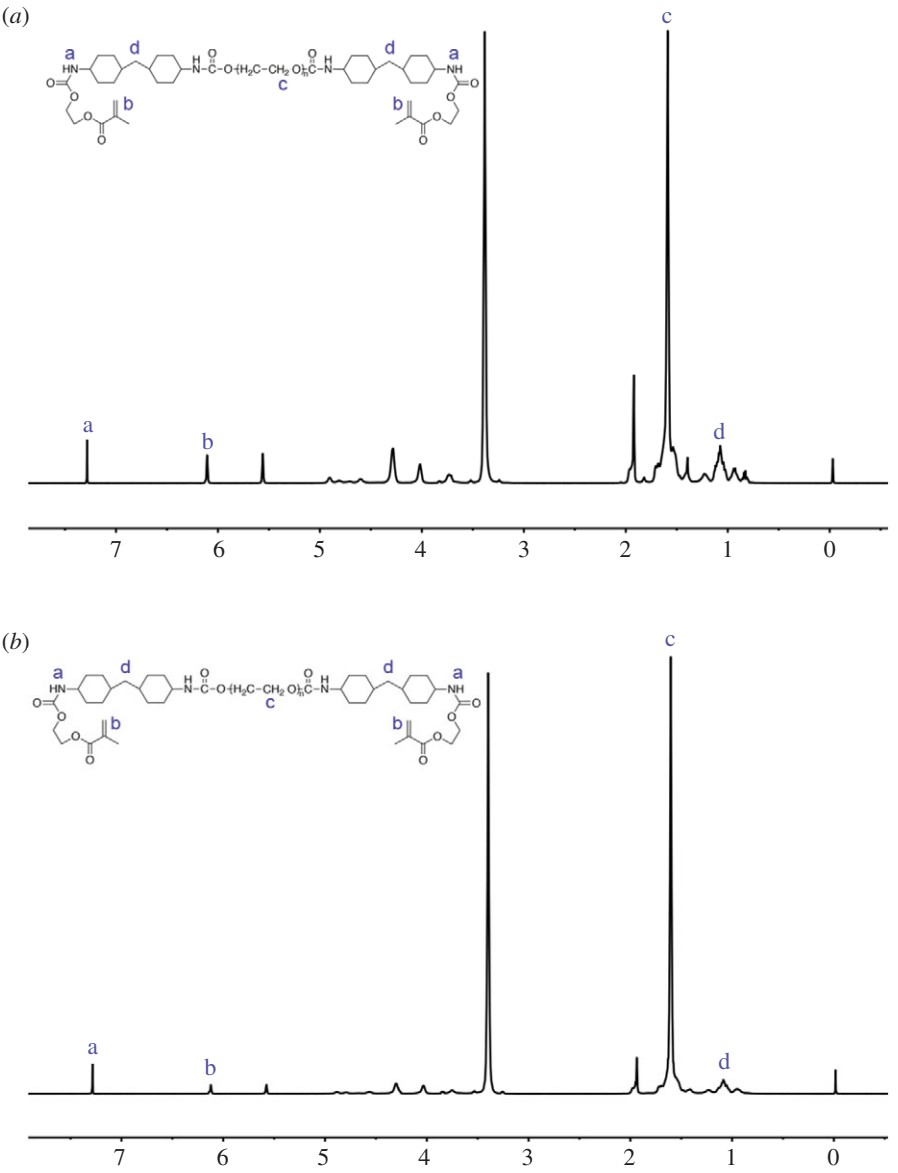

**Figure 3.** (a) ¹H NMR spectra of PUA. (b) ¹H NMR spectra of PUB.

the absorption of water, which may affect the dispersion of the interfacial stress and eventually lead to interfacial damage after repeated loadings.

Figure 5d shows that the water absorption values of the seven PU adhesives are significantly lower than those of the three commercial adhesives ($p < 0.05$). PU adhesives are mainly composed of oligomers, containing urethane groups that are relatively hydrophobic and may form weaker hydrogen bonds with water molecules when compared to hydroxyl groups, reflected by lower cohesive energy density (urethane group: $1425 \, J \, cm^{-3}$; OH group: $2980 \, J \, cm^{-3}$) [32]. This is consistent with the results of the static contact angle, as shown in figure 5e. The contact angles of the PU adhesives are all greater than 83°, significantly larger than that of the commercial adhesive ($p < 0.05$), indicating that the PU adhesive is more hydrophobic than the commercial adhesive.

After the polymer absorbs moisture, the polymer network is softened by swelling itself and reducing the friction between the polymer chains [33]. When too much moisture is absorbed, the macromolecular polymer chains undergo a relaxation process. Meanwhile, the residual monomer in the polymer is released to the surrounding environment at a rate that is related to the swelling and relaxation ability of the polymer. A more hydrophilic polymer network, such as commercial adhesives, with better relaxation capacity allows for faster release of residual monomers through the nanovoids in the material [34,35], resulting in a decreasing quality under short-term water soaking. At the same time,

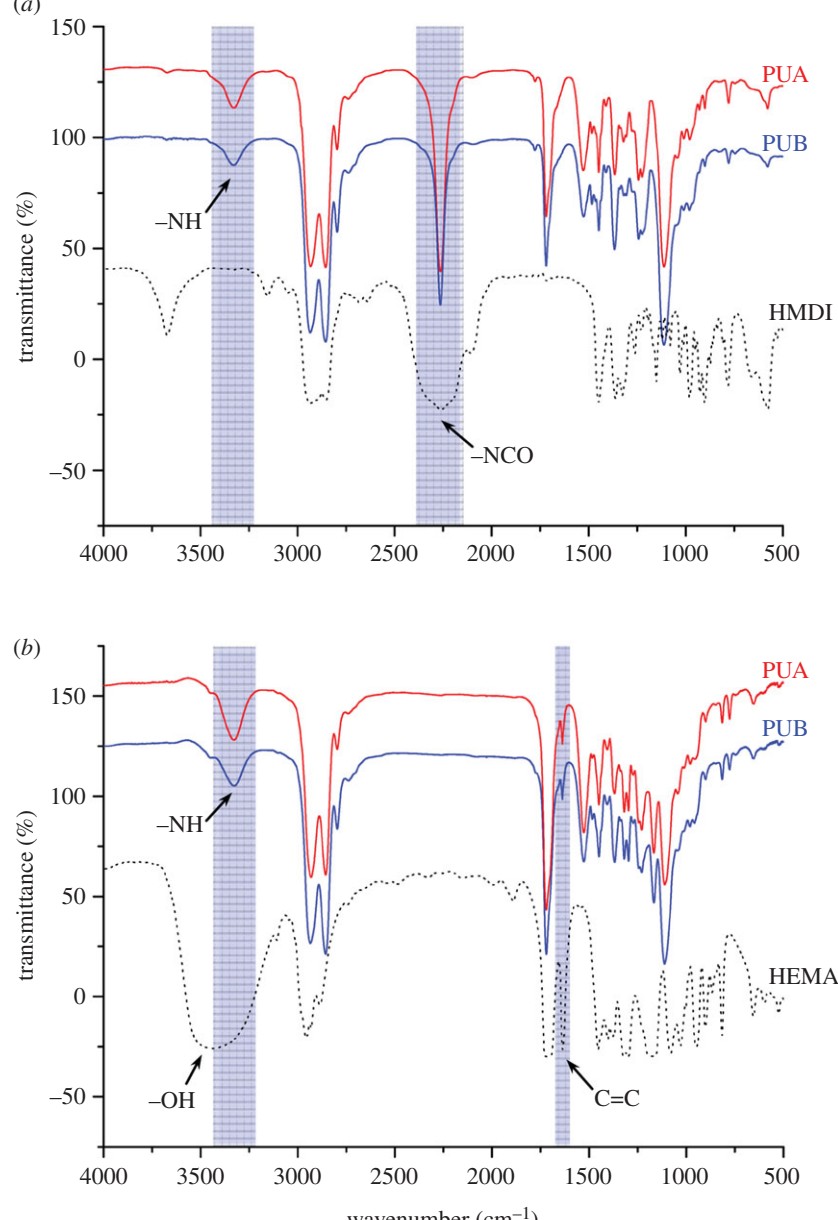

**Figure 4.** (*a*) FTIR spectra of the NCO-terminated PU prepolymer. (*b*) FTIR spectra of the PU oligomer.

the released residual monomers, such as TEGDMA and HEMA, enter the dentinal tubules, causing harmful inflammatory responses [36,37]. The components dissolved from the adhesives have a potential negative effect on the stability of their own structures, ultimately resulting in degradation of the resin–dentine bonding interface. Compared with these methacrylic resin adhesives, the solubility of PU adhesives was significantly reduced ($p < 0.05$), as shown in figure 5*c*. The soft segment of the synthetic PU adhesives (polytetrahydrofuran diol) imparts elasticity and hydrolysis resistance to the adhesives. The amount of released residual monomers is also dependent on the DC of the monomers [32]. The PU adhesive with a higher DC showed a lower amount of residual monomer release, improving the bonding durability.

## 3.3. Tensile strength and elongation at break

The tensile strength and elongation at break is mainly used for testing materials with elasticity, such as the synthesized PU adhesive in this study. The composition of the commercial adhesive system is different from that of the experimental PU adhesive. The commercial adhesive is mainly composed of the organic resin matrix and inorganic filler, which is a rigid structure with almost no tensile

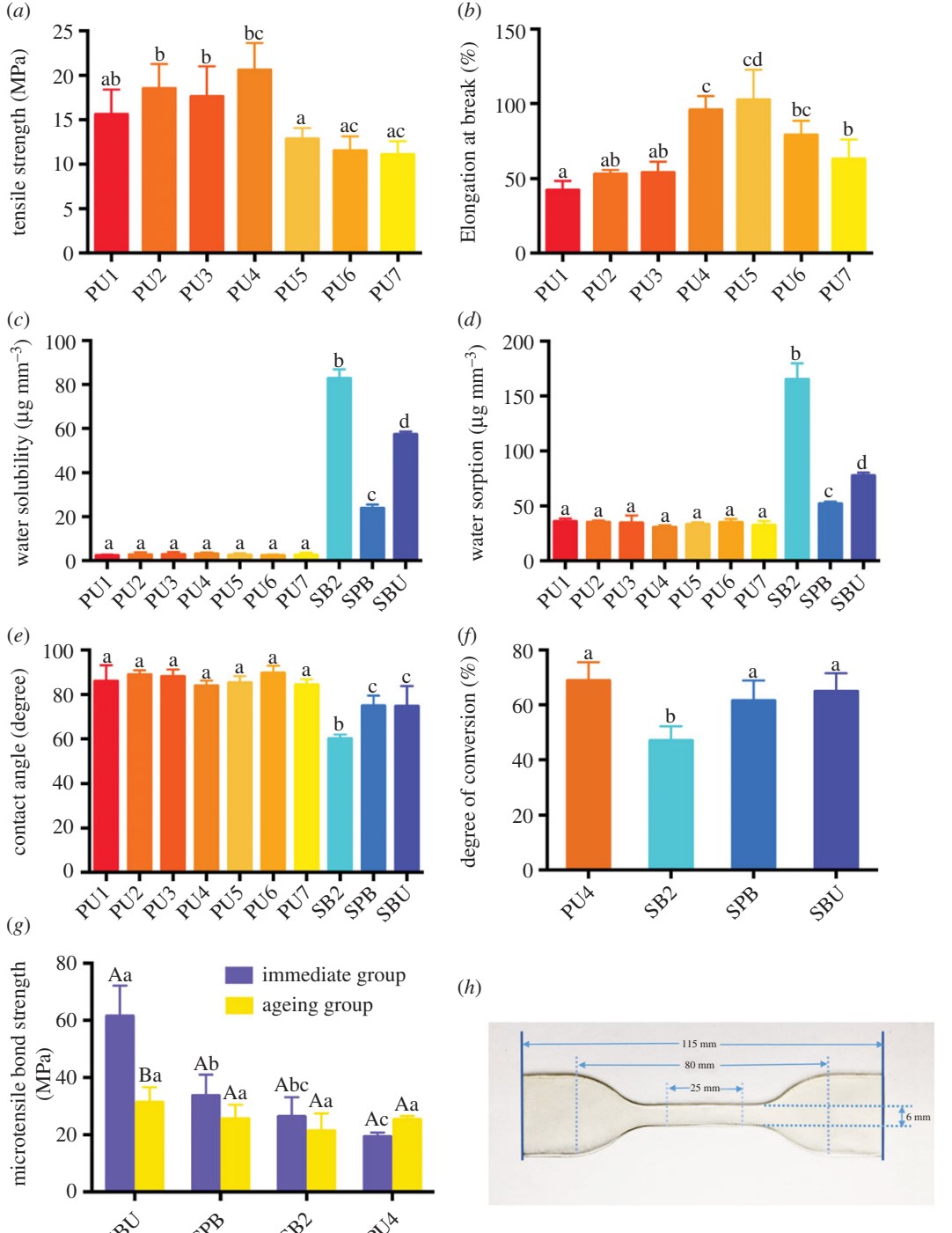

**Figure 5.** (*a,b*) Tensile strength and elongation at break of PU adhesives. (*c,d*) Water solubility and water sorption of PU adhesives. (*e*) Contact angle of PU adhesives. (*f*) Degree of conversion of commercial adhesives and PU4 (different upper-case letters indicate the difference ($p < 0.05$) between the immediate group and ageing group of each adhesive; different lower-case letters represent the difference ($p < 0.05$) among adhesives in either the immediate group or the ageing group). (*g*) Microtensile bond strength of commercial adhesives and PU4 adhesive before and after ageing. (*h*) Photograph of the specimen for tensile property testing.

deformation. Therefore, commercial adhesives were not tested. Figure 5*a* showed that there was no significant difference among PU2, PU3 and PU4, but their tensile strength was significantly higher than that of other groups ($p < 0.05$). In terms of elongation at break, as shown in figure 5*b*, there was no significant difference between PU4 and PU5, and their elongation was obviously higher than that of the other groups ($p < 0.05$). The mechanical properties of PU adhesives depend on the soft segment and hard segment [38]. The soft segment used in this study is polytetrahydrofuran1000/2000, which

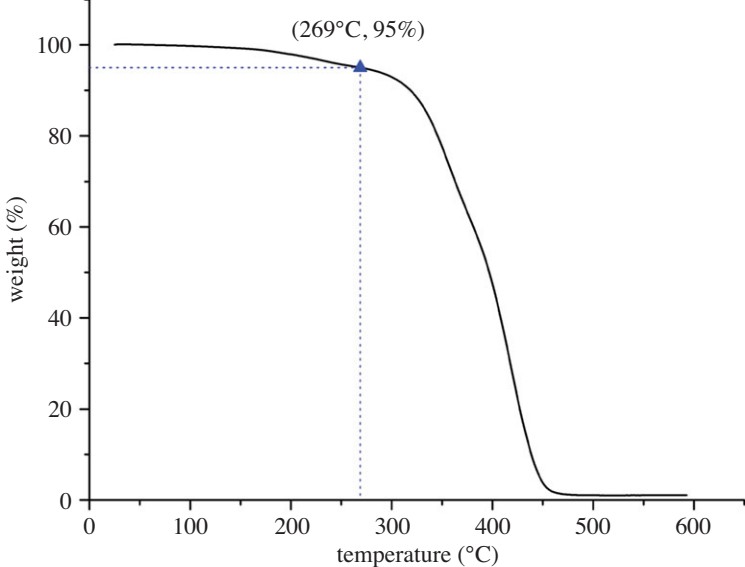

**Figure 6.** TGA spectrum of PU4 adhesive.

endows synthesized PU adhesives with good mechanical properties, flexibility and hydrolysis resistance [39]. In PU1–PU3, PUA synthesized by PTMG1000 accounts for a larger proportion, leading to a relatively high amount of hard segments in PU adhesives. Therefore, PU1–PU3 has better mechanical properties. The PUB synthesized by PTMEG2000 occupies a larger percentage in PU5–PU7, resulting in a relatively large amount of soft segments in PU adhesives. Therefore, PU5–PU7 has better elastic properties and greater elongation at break, but after reaching a certain extent, the elongation at break is lowered due to the decreasing mechanical properties. Among these seven adhesives, PU4 possesses the highest tensile strength and relatively higher elongation at break. Considering the water absorption/solubility and contact angle of seven kinds of PU adhesives comprehensively, PU4 was considered to have the best performance and chosen as the final experimental adhesive formulation for the follow-up dental restoration tests to compare with three commercial adhesives.

During the cross-linking of the polymer initiated by light irradiation, radical polymerization occurs, and the polymer chain becomes denser, resulting in a decrease in volume (average 1% to 3%) [40]. Polymerization shrinkage causes internal stress, leading to pain, microleakage and secondary caries, and ultimately failure of the restoration [41,42]. The PU elastic adhesive can buffer stress coming from polymerization shrinkage, inconsistent thermal expansion coefficients between dentine and composite resin and occlusal force by deformation. Elongation at the break of PU4 reached 95.74%; according to Yingchao *et al.*'s report, this can meet the requirements [13]. Therefore, the PU4 adhesive can reduce the interfacial stress, decrease the occurrence of microleakage and secondary caries, and improve the stability and durability of the adhesive.

## 3.4. Thermal stability characterization

Assessment of the thermal properties of the PU4 adhesive is important to assess its applicability in the oral environment. TGA was carried out to investigate the thermal performance of the PU4 adhesive. Figure 6 shows the TGA thermogram results of the PU4 adhesive. The initial degradation temperature of 5% weight loss was observed at 269.00°C. The maximum tolerant temperature of the oral mucosa is approximately 60°C. Therefore, PU4 adhesive can be applied to the oral environment. And more thermal properties may be investigated in the future, such as DSC.

## 3.5. Degree of conversion

Adequate polymerization of the adhesive layer is necessary to ensure its physical, chemical and mechanical strength. The DC of the dental adhesive is closely related to the structure of the monomers, the polymerization conditions and the photoinitiator concentration [43]. PU4 adhesive is mainly composed of oligomers. As shown in figure 5f, the DC of PU4 reached 68.86 ± 6.72%, which

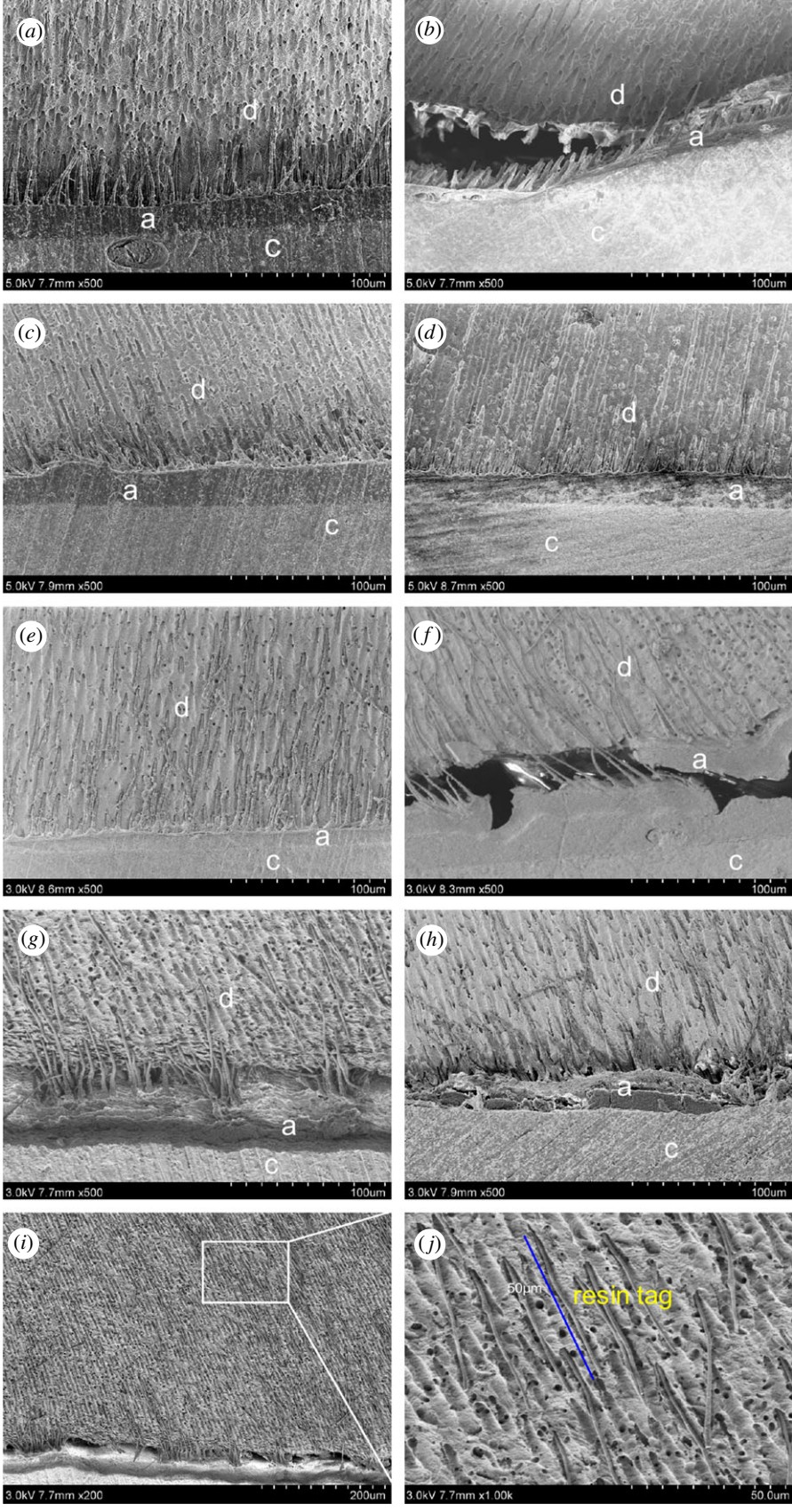

**Figure 7.** SEM of the bonding surface of commercial adhesives and PU4 adhesive before and after 5000 thermocyclings. (*a*) SB2, immediate (*b*) SB2, ageing (*c*) SPB, immediate (*d*) SPB, ageing (*e*) SBU, immediate (*f*) SBU, ageing (*g*) PU4 immediate (*h*) PU4, ageing (*i,j*) PU4, immediate. PU4 adhesive can penetrate into the dentinal tubules, some of which even reach 50 μm. (d, dentine; c, composite resin; a, adhesive layer.)

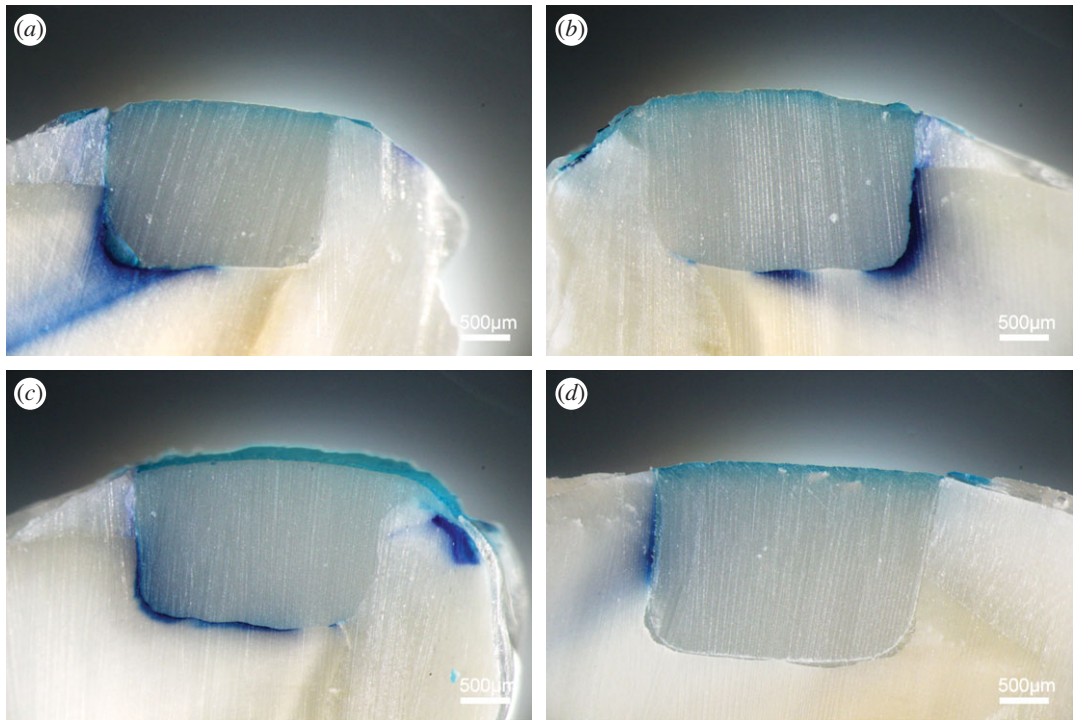

**Figure 8.** Microleakage between dentine and composite resins after 5000 thermocycling. (*a*) SB2; (*b*) SPB; (*c*) SBU; (*d*) PU4. Scale bar = 500 μm.

was higher than that of the commercial adhesive SB2 ($p = 0.0004$). A higher DC of the adhesive is advantageous for the improvement of the bonding quality, because it may enhance the strength of the adhesive layer, thus improving the bonding durability.

## 3.6. Microtensile bond strength test

Figure 5*g* shows the microtensile strength of PU4 adhesive and three commercial adhesives before and after 5000 thermocycles. In the immediate group, the microtensile strength of the PU4 adhesive was lower than that of the three commercial adhesives. This is due to the elastic properties of the PU4 adhesive and its cohesive energy is lower than that of the rigid commercial adhesives. Although the immediate microtensile strength of PU4 is low, it is also greater than 20 MPa, which can meet clinical requirements [13]. After ageing, the microtensile strength of the three commercial adhesives decreased, and the SBU group showed the most obvious decline. However, the microtensile strength of the PU4 adhesive increased. On the one hand, PU4 adhesive can buffer the stress during 5000 thermocyclings. On the other hand, the urethane group (-NHCOO-) contained in the PU4 adhesive has a strong polarity, and hydrogen bonds can be formed between the adhesive molecules and between the adhesive and the dentine, thereby enhancing the molecular cohesion and improving the bonding strength and durability. In term of the microtensile experiment, SBU is better than PU4, but water absorption and water solubility of PU4 are much lower than that of SBU which may contribute to reducing hydrolytic degradation of the bonding interface. And microleakage of PU4 has also been improved when compared with SBU. Its elastic properties can buffer various stresses during long-term use, improving the stability and durability of the adhesive bonding interface. Therefore, the comprehensive performance of PU4 is still better than that of SBU.

The mechanism of dentine bonding allows the adhesive monomer to penetrate into the demineralized dentine collagen fibre matrix to form a hybrid layer (HL), resulting in micromechanical interlocking [4,6]. The bonding interface of the adhesive was detected by scanning electron microscopy, as shown in figure 7. A good bonding interface can be observed in all the immediate groups. PU4 adhesive can also penetrate into the dentinal tubules, some of which even reach 50 μm in depth. After 5000 thermocyclings, significant cracks can be seen in the bonding interface of SB2 and SBU. However, the hybrid layer of the PU4 adhesive remains intact.

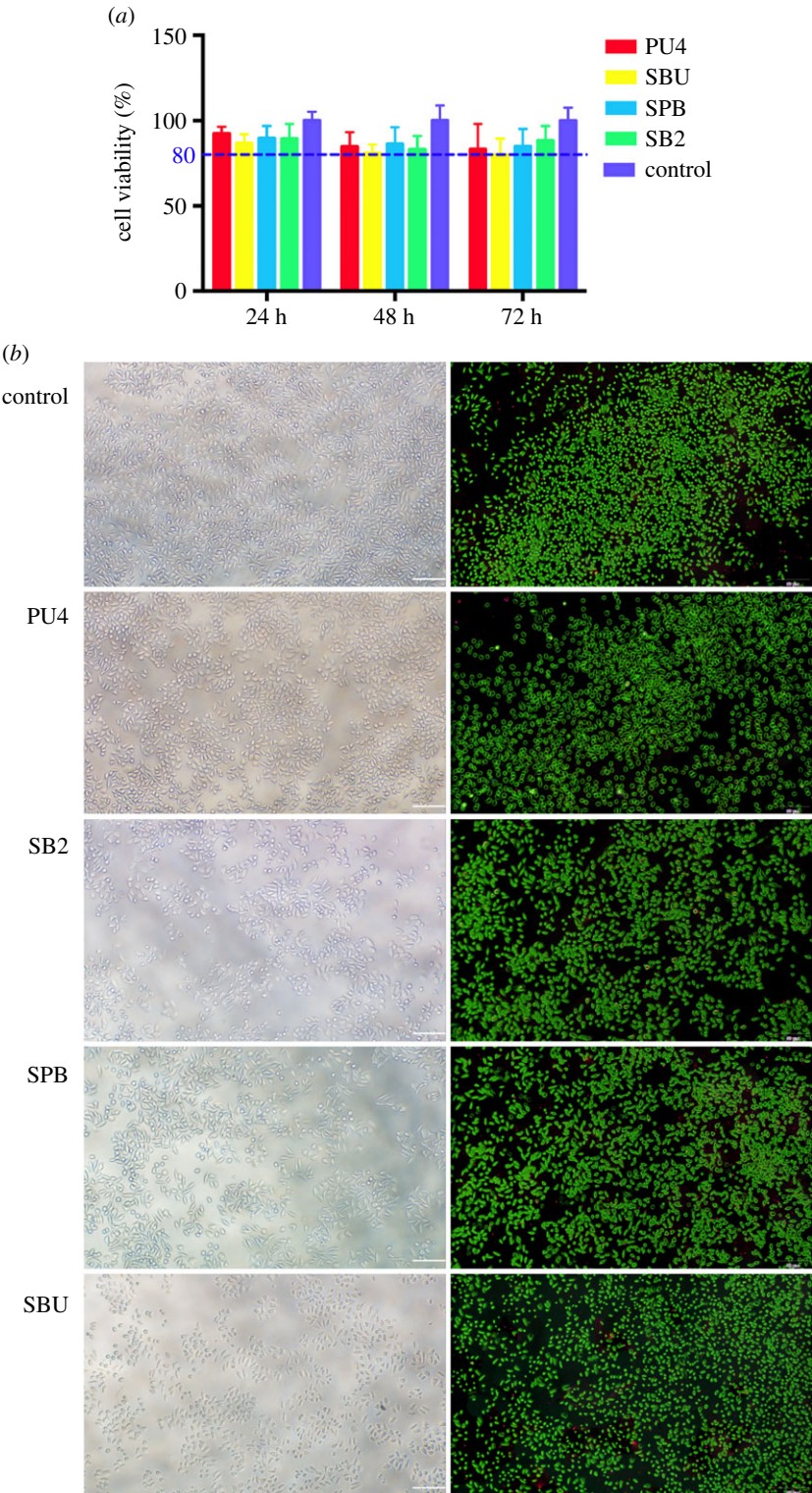

**Figure 9.** (*a*) Cytotoxicity of the PU4 adhesive and commercial adhesives SB2, SPB and SBU was determined by MTT assay. A cell relative survival rate greater than 75% can be considered non-cytotoxic. (*b*) Optical microscopy images (left) and live/dead fluorescence cell staining (right) after L929 cells were cultured for 24 h in culture medium (control), extracted liquid of the PU4 adhesive and commercial dental adhesives SB2, SPB and SBU. Living cells were stained with calcium—AM (green), and dead cells were stained with PI (red). Scale bars = 100 μm.

## 3.7. Microleakage in composite restoration

The microleakage of group SB2, group SPB and group SBU reached the axial surface or even the pulp with depth of $1.75 \pm 0.08$, $1.79 \pm 0.10$, and $1.70 \pm 0.12$ mm, respectively (grade 3/4). The microleakage

of group PU4 exceeded half of the depth of the hole but did not involve the axial surface and only reached $0.98 \pm 0.16$ mm (grade 2), as shown in figure 8. There is no significant difference among group SB2, group SPB and group SBU. However, the microleakage of commercial adhesive is significantly higher than that of PU4 adhesive ($p < 0.0001$). Thermocycling is often used in dental adhesive experiments to simulate temperature changes in the oral cavity. Due to the difference in thermal expansion coefficient between the composite resin and the tooth, stress will be generated after repeated temperature changes, which eventually accelerates the formation of microleakage [44]. In this study, PU4 adhesive can significantly reduce the occurrence of microleakage, mainly due to its elastic properties, which change the rigid connection to elastic bonding between composite resin and dentine. The stress generated by the inconsistent thermal expansion coefficient can be buffered through deformation of the PU4 adhesive, reducing the probability of fracture and secondary caries and improving the stability and durability of the adhesive. The microleakage of the small molecule dye is qualitative, so more evaluation is needed in the future study.

## 3.8. Biocompatibility of polyurethane adhesive

Cytotoxicity was evaluated by examining the effect of PU4 adhesive on cell activity and morphological changes in L929 fibroblasts. As shown in figure 9a, there was no significant difference between the experimental groups and the blank control group in cell proliferation activity, regardless of the time (24 h, 48 h, 72 h) of extraction. According to GB/T 16886.5–2003 (ISO 10993–5:1999), samples with cell viability larger than 75% of blank group can be considered as non-cytotoxic [22]. In this experiment, the cell viability was all greater than 80%. It is important to mention that the commercial adhesive system may show some toxicity to a certain extent. Adhesive systems components (e.g. Bis-GMA, HEMA, TEGDMA, UDMA and camphorquinone) are cytotoxic. Despite its cytotoxicity produced by direct contact on pulp cells, these may be clinically attenuated or neutralized, as adhesive systems are applied to dentine, which acts as a physical barrier depending on its thickness.

L929 cells were cultured for 24 h using the 24 h extraction, and their cell viability was quantified by a live/dead staining experiment. As shown in figure 9b, the proportion of living cells was calculated using ImageJ software, and the result is consistent with that of the MTT experiment. The morphology of L929 cells was directly observed with an inverted microscope after culture with the 24 h extraction, as shown in figure 9b. L929 cells were lengthened and became spindle-shaped, similar to the morphology of typical fibroblasts in the control group. The cell culture was evenly distributed, and the intercellular spaces did not change significantly. This indicates that PU4 adhesive can meet the clinical biosafety demands.

# 4. Conclusion

In summary, an elastic PU adhesive was prepared and evaluated using comprehensive methods. The lower water sorption/solubility and decreased microleakage of PU adhesive indicates that it can prevent water from permeating into the bonding interface and enhance marginal sealing. The PU adhesive can also buffer the stress coming from volumetric polymerization shrinkage, temperature changes and repeated chewing force by deformation due to its elastic property. Furthermore, it is biosafe for L929 cells. This study lays the foundation for the application of PU adhesive in clinical practice to produce stable and long-lasting adhesion.

Ethics. All relevant ethical applications in this experiment were approved by the Ethics Committee of School and Hospital of Stomatology of Jilin University. Informed consent was obtained from all donors.
Data accessibility. Data are available in the electronic supplementary material.
Authors' contributions. The authors meet all of the following criteria:

1) substantial contributions to conception and design, or acquisition of data, or analysis and interpretation of data;
2) drafting the article or revising it critically for important intellectual content;
3) final approval of the version to be published; and
4) agreement to be accountable for all aspects of the work in ensuring that questions related to the accuracy or integrity of any part of the work are appropriately investigated and resolved.

J.Zhang conducted the statistical analysis and wrote this manuscript; X.G. and X.Z. finished the acquisition of data; H.W., J.Zhu and Z.S. conducted the experiments and repeated them; S.Z. and Z.C. designed and oversaw the study, and reviewed and revised this manuscript.

Competing interests. We declare we have no competing interests.

Funding. This research was financially supported by the National Natural Science Foundation of China (NSFC, grant no. 81671033).

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
