## [Reviewer comments · Royal Society Open Science]

Review History

RSOS-200457.R0 (Original submission)

Review form: Reviewer 1

Is the manuscript scientifically sound in its present form?

Yes

Are the interpretations and conclusions justified by the results?

Yes

Is the language acceptable?

Yes

Do you have any ethical concerns with this paper?

No

Have you any concerns about statistical analyses in this paper?

Yes

Recommendation?

Major revision is needed (please make suggestions in comments)

Comments to the Author(s)

The paper is in general well written and describes a clinically relevant topic. However, several aspects needs to be answered.

Abstract.

It's necessary that all methodologies performed are specified.

The conclusion does not answer the objective of the paper.

Fig.1 and 3 are unnecessary. Graphic abstract is more suitable.

Some methodologies used are insufficient described, as contact angle (how the calculation was performed; kind of liquid; device information; software used); thermal stability characterization; failure pattern in SEM.

Which light-curing unit was used?

Only PU4 was used as experimental group for several methodologies. Clarify.

Statistical test of normality and homoscedasticity distribution is required.

Only two-way ANOVA is presented for Microtensile bond strength with uncertainty post-hoc test (Tukey or Dunnet?). It is necessary to present all statistical analyzes for all the methodologies developed.

In the results topic, p-values are presented ($p < 0.05$). It is required to present the specific p-value, when p is statistically significant.

Letters indicating statistical difference in alphabetical order relating lower to higher means are preferable.

Was microleakage qualitatively evaluated?

Z350XT is a conventional resin composite used as oblique and incremental technique. However, the author used this material as bulk fill, with 4-5 increment layer. This technique directly influences the adhesive performance, compromising the results and discussion.

I suggest that it be redone correctly (without research bias) or removed.

The thickness of resin composite directly influences on behavior of composite material, as well as the interface bonding. Therefore, it can be observed that fig. 9C presents a greater thickness compared to other images, influencing marginal microleakage in the back wall of the cavity (bias). Clarify.

Review form: Reviewer 2

Is the manuscript scientifically sound in its present form?

Yes

Are the interpretations and conclusions justified by the results?

No

Is the language acceptable?

Yes

Do you have any ethical concerns with this paper?

No

Have you any concerns about statistical analyses in this paper?

Yes

Recommendation?

Major revision is needed (please make suggestions in comments)

Comments to the Author(s)

Journal: Royal Society Open Science

Manuscript ID: RSOS-200457

Title: Hydrolysis-resistant and stress-buffering bifunctional polyurethane adhesive for durable dental composite restoration.

In this manuscript a new elastic polyurethane (PU) adhesive is reported. After its synthesis by the solution polymerization method, it was evaluated in different endpoints: water sorption, water solubility, contact angle, thermal stability, mechanical properties, and in-vitro biocompatibility. This new material has better stability and durability of the dental adhesion interface in comparison to other commercial.

Comments:

- 1) Table 1 should be simplified. For example, the manufacturer's name could be removed, and the batch number could go as a table foot.
- 2) Figure 6A and 6B show the mechanical properties of the synthesized PUs. However, these properties of commercial PUs are not shown. These data are also not included in the discussion of the results.
- 3) Why are the conversion data for samples PU1, PU2, PU3, PU5 and PU6 not included?
- 4) The microtensile bond strength can be related to tensile strength?. Explain it. By other hand, both microtensile and tensile are measured in the same scale (MPa)?
- 5) In the TGA curve there is a small inflection in the slope, if the curve was derived, it could detect PUA and PUB ?. Could you show DSC of PUA, PUB and PU4?.
- 6) Figure 8F shows that SBU aging can also penetrate the dentinal tubules, and Figure 6G shows that the microtensile bond strength of SBU is greater than PU4 ?. Could you say that by this measure that PU4 is not better than SBU?.
- 7) In Figure 10A, the viability values are not shown in percentages as indicated (i.e., 1.0= 100%). Setting viability at 70% is arbitrary (the reader can suppose a 30% death of fibroblasts in the mouth?). What is the variation in your controls?. Even a 20% decrease in viability, this is significant in a cell population?. You can perform an ANOVA followed by a Tukey test to set significance to 5%.
- 8) The light microscopy images in Figure 10B must be improved, or show SEM images, in order to visualize the morphology of the cells. With the viability shown in Figure 10A, some dead (red) cells should be observed. Fibroblasts grow extended (in spreading), however the fluorescence image gives the appearance of cluster or aggregates. Can you explain it?

Decision letter (RSOS-200457.R0)

Dear Dr Zhang:

Title: Hydrolysis-resistant and stress-buffering bifunctional polyurethane adhesive for durable dental composite restoration
Manuscript ID: RSOS-200457

The editor assigned to your manuscript has now received comments from reviewers. We would like you to revise your paper in accordance with the referee and Subject Editor suggestions which can be found below (not including confidential reports to the Editor). Please note this decision does not guarantee eventual acceptance.

Please submit your revised paper before 22-May-2020. Please note that the revision deadline will expire at 00.00am on this date. If we do not hear from you within this time then it will be assumed that the paper has been withdrawn. In exceptional circumstances, extensions may be possible if agreed with the Editorial Office in advance. We do not allow multiple rounds of revision so we urge you to make every effort to fully address all of the comments at this stage. If deemed necessary by the Editors, your manuscript will be sent back to one or more of the original reviewers for assessment. If the original reviewers are not available we may invite new reviewers.

RSC Associate Editor:
Comments to the Author:
(There are no comments.)

RSC Subject Editor:
Comments to the Author:
(There are no comments.)

Reviewers' Comments to Author:
Reviewer: 1

Comments to the Author(s)
The paper is in general well written and describes a clinically relevant topic. However, several aspects needs to be answered.

Abstract.
It's necessary that all methodologies performed are specified.
The conclusion does not answer the objective of the paper.

Fig.1 and 3 are unnecessary. Graphic abstract is more suitable.

Some methodologies used are insufficient described, as contact angle (how the calculation was performed; kind of liquid; device information; software used); thermal stability characterization; failure pattern in SEM.
Which light-curing unit was used?
Only PU4 was used as experimental group for several methodologies. Clarify.

Statistical test of normality and homoscedasticity distribution is required.
Only two-way ANOVA is presented for Microtensile bond strength with uncertainty post-hoc test (Tukey or Dunnet?). It is necessary to present all statistical analyzes for all the methodologies developed.
In the results topic, p-values are presented ($p < 0.05$). It is required to present the specific p-value, when p is statistically significant.
Letters indicating statistical difference in alphabetical order relating lower to higher means are preferable.
Was microleakage qualitatively evaluated?

Z350XT is a conventional resin composite used as oblique and incremental technique. However, the author used this material as bulk fill, with 4-5 increment layer. This technique directly influences the adhesive performance, compromising the results and discussion.
I suggest that it be redone correctly (without research bias) or removed.

The thickness of resin composite directly influences on behavior of composite material, as well as the interface bonding. Therefore, it can be observed that fig. 9C presents a greater thickness compared to other images, influencing marginal microleakage in the back wall of the cavity (bias). Clarify.

Reviewer: 2

Comments to the Author(s)

Journal: Royal Society Open Science

Manuscript ID: RSOS-200457

Title: Hydrolysis-resistant and stress-buffering bifunctional polyurethane adhesive for durable dental composite restoration.

In this manuscript a new elastic polyurethane (PU) adhesive is reported. After its synthesis by the solution polymerization method, it was evaluated in different endpoints: water sorption, water solubility, contact angle, thermal stability, mechanical properties, and in-vitro biocompatibility. This new material has better stability and durability of the dental adhesion interface in comparison to other commercial.

Comments:

- 1) Table 1 should be simplified. For example, the manufacturer's name could be removed, and the batch number could go as a table foot.
- 2) Figure 6A and 6B show the mechanical properties of the synthesized PUs. However, these properties of commercial PUs are not shown. These data are also not included in the discussion of the results.
- 3) Why are the conversion data for samples PU1, PU2, PU3, PU5 and PU6 not included?
- 4) The microtensile bond strength can be related to tensile strength?. Explain it. By other hand, both microtensile and tensile are measured in the same scale (MPa)?
- 5) In the TGA curve there is a small inflection in the slope, if the curve was derived, it could detect PUA and PUB ?. Could you show DSC of PUA, PUB and PU4?.
- 6) Figure 8F shows that SBU aging can also penetrate the dentinal tubules, and Figure 6G shows that the microtensile bond strength of SBU is greater than PU4 ?. Could you say that by this measure that PU4 is not better than SBU?.
- 7) In Figure 10A, the viability values are not shown in percentages as indicated (i.e., 1.0= 100%). Setting viability at 70% is arbitrary (the reader can suppose a 30% death of fibroblasts in the mouth?). What is the variation in your controls?. Even a 20% decrease in viability, this is significant in a cell population?. You can perform an ANOVA followed by a Tukey test to set significance to 5%.
- 8) The light microscopy images in Figure 10B must be improved, or show SEM images, in order to visualize the morphology of the cells. With the viability shown in Figure 10A, some dead (red) cells should be observed. Fibroblasts grow extended (in spreading), however the fluorescence image gives the appearance of cluster or aggregates. Can you explain it?

Author's Response to Decision Letter for (RSOS-200457.R0)

See Appendix A.

RSOS-200457.R1 (Revision)

Review form: Reviewer 2

Is the manuscript scientifically sound in its present form?

Yes

Are the interpretations and conclusions justified by the results?

Yes

Is the language acceptable?

Yes

Do you have any ethical concerns with this paper?

No

Have you any concerns about statistical analyses in this paper?

No

Recommendation?

Accept with minor revision (please list in comments)

Comments to the Author(s)

Many of the responses to the reviewer are information that the reader needs to understand the manuscript, especially the parts marked with yellow (see Appendix B). Authors have to include and/or adapt these parts in the paragraphs of their main text for better clarity of the paper.

Decision letter (RSOS-200457.R1)

Dear Dr Zhang:

Title: Hydrolysis-resistant and stress-buffering bifunctional polyurethane adhesive for durable dental composite restoration

Manuscript ID: RSOS-200457.R1

Thank you for submitting the above manuscript to Royal Society Open Science. On behalf of the Editors and the Royal Society of Chemistry, I am pleased to inform you that your manuscript will be accepted for publication in Royal Society Open Science subject to minor revision in accordance with the referee suggestions. Please find the reviewers' comments at the end of this email.

The reviewers and handling editors have recommended publication, but also suggest some minor revisions to your manuscript. Therefore, I invite you to respond to the comments and revise your manuscript.

Because the schedule for publication is very tight, it is a condition of publication that you submit the revised version of your manuscript before 13-Jun-2020. Please note that the revision deadline

will expire at 00.00am on this date. If you do not think you will be able to meet this date please let me know immediately.

Kind regards,

Dr Laura Smith
Publishing Editor, Journals

RSC Associate Editor:
Comments to the Author:
(There are no comments.)

RSC Subject Editor:
Comments to the Author:
(There are no comments.)

Reviewer comments to Author:
Reviewer: 2

Comments to the Author(s)
Many of the responses to the reviewer are information that the reader needs to understand the manuscript, especially the parts marked with yellow (see attach file). Authors have to include and/or adapt these parts in the paragraphs of their main text for better clarity of the paper.

Author's Response to Decision Letter for (RSOS-200457.R1)

See Appendix C.

Decision letter (RSOS-200457.R2)

Dear Dr Zhang:

Title: Hydrolysis-resistant and stress-buffering bifunctional polyurethane adhesive for durable dental composite restoration
Manuscript ID: RSOS-200457.R2

It is a pleasure to accept your manuscript in its current form for publication in Royal Society Open Science. The chemistry content of Royal Society Open Science is published in collaboration with the Royal Society of Chemistry.

RSC Associate Editor
Comments to the Author:
(There are no comments.)

Reviewer(s)' Comments to Author:

Appendix A

Point by point response to the reviewers' comments

Dear Editor and Reviewers,

We hope you are keeping well at this difficult time.

Thanks very much for taking your time to review this manuscript. I really appreciate all these precious comments and suggestions. Please find my itemized responses in below and my revisions in the resubmitted files.

Thanks again.

Reviewer: 1

1) Abstract.

It's necessary that all methodologies performed are specified.

The conclusion does not answer the objective of the paper.

We are grateful for the suggestion.

We have revised the abstract according to the suggestion that all methodologies performed are specified and the conclusion answers the objective of the paper. The revised text is as following. "A new elastic polyurethane (PU) adhesive was reported in this study to improve the stability and durability of the dental adhesion interface. A polyurethane oligomer was synthesized by the solution polymerization method, and a diluent and solvent were added to prepare PU adhesives. The water sorption, water solubility, contact angle, thermal stability, **degree of conversion** and mechanical properties of the PU adhesives were evaluated. Experimental applications for tooth restoration (**microtensile bond strength and microleakage**) were also performed. **And cytotoxicity test was carried out.** The water sorption and solubility of the PU adhesives were significantly lower than those of three commercial adhesives. The microtensile bond strength of the PU adhesives was improved after thermocycling test, and the extent of microleakage was diminished when compared with that of commercial adhesives. Biocompatibility testing demonstrated that the PU adhesive was nontoxic to L929 fibroblasts. This study **shows the ability of PU adhesive to improve the stability and durability of the dental adhesion interface** and may refocus the attention of scientists from rigid bonding to flexible bonding for dental adhesion, and it sheds light on a new strategy for the stable and durable bonding interface of dentin adhesives."

2) Fig.1 and 3 are unnecessary. Graphic abstract is more suitable.

We are grateful for the suggestion.

Based on the comments, Fig.1 and 3 were removed in the article and graphic abstract was added.

3) Some methodologies used are insufficient described, as contact angle (how the calculation was performed; kind of liquid; device information; software used); thermal stability characterization; failure pattern in SEM.

We are grateful for the suggestion.

More detailed information was added to supplement the insufficient methodologies according to the comment to make it more clearly. The revised text is as following.

contact angle

“Contact angles were obtained using the sessile drop method with a Dataphysics contact angle analyzer (OCA-20, DataPhysics Co., German). This instrument consists of a CCD video camera with a resolution of 768×576 pixel and up to 50 images per second and multiple microsyringe units. A drop of $6 \mu\text{l}$ of deionized water was gently dropped onto the surface of the adhesive to take a digital photo. The digital drop image was processed by a specialized software SCA 20, which calculated both the left and right contact angles from the shape of the drop.” The representative digital drop images are shown in the following picture.

thermal stability characterization;

“Thermogravimetric analysis measurement of PU4 adhesive was performed using a TGA thermal analyzer (Q500, TA Instruments, USA). Initial sample masse is around 5 mg. The heating rate is $10 \text{ }^\circ\text{C}/\text{min}$. The experiments were performed in an inert atmosphere with a continuous flow of nitrogen at the rate of $150 \text{ ml}/\text{min}$ and heated from room temperature to $800 \text{ }^\circ\text{C}$.”

failure pattern in SEM.

The SEM in this study is the intact bonding surface of the PU4 adhesive and three commercial adhesives. We are sorry that failure pattern in SEM was not conducted in this study. In the following study, we will supplement this experiment according to the suggestion.

Which light-curing unit was used?

The light-curing unit was a LED light (SLC-VIIIA, China) which was used in clinical practice. The LED light had an output light intensity of approximately 900 mW/cm², which was monitored by a radiometer to ensure light intensity.

Only PU4 was used as experimental group for several methodologies. Clarify.

In this study, we prepared seven kinds of PU adhesives. Then, we conduct several experiments (tensile strength, elongation at break, water solubility, water sorption and contact angle) to evaluate them. Finally, among these seven adhesives, PU4 possesses the highest tensile strength and relatively higher elongation at break. Considering the water absorption/solubility and contact angle of seven kinds of PU adhesives comprehensively, PU4 was considered to have the best comprehensive performance and chosen as the final experimental adhesive formulation for the following tests (degree of conversion, microtensile bond strength, microleakage and biocompatibility) to compare with three commercial adhesives. So, only PU4 was used as experimental group for the following methodologies.

4) Statistical test of normality and homoscedasticity distribution is required. Only two-way ANOVA is presented for Microtensile bond strength with uncertainty post-hoc test (Tukey or Dunnet?). It is necessary to present all statistical analyzes for all the methodologies developed.

We are grateful for the suggestion.

Statistical test of normality and homoscedasticity distribution has been conducted. All statistical analyzes for all the methodologies developed have been presented in the section of statistical analysis. We have added this section in the text of statistical analysis to make it more clearly. The revised text is as following.

“Data were expressed as the mean \pm standard deviation. The data were consistent with normality and homoscedasticity distribution. Data for microtensile bond strength were analyzed using two-

way ANOVA, and the data of tensile strength, elongation at break, water solubility, water sorption, contact angle and degree of conversion were submitted to one-way ANOVA using SPSS software (version 19.0, SPSS Inc., Chicago, IL, USA). Multiple comparison analysis was conducted using the Tukey test. The significance level was set at $p = 0.05$ for this study.”

In the results topic, p-values are presented ($p < 0.05$). It is required to present the specific p-value, when p is statistically significant.

The specific p-values have been presented when p is statistically significant in the manuscript. However, there are several summary statements with more than one P value. Therefore, we have listed them in the following tables to ensure the conciseness of the article.

(1) “Compared with these methacrylic resin adhesives, the solubility of PU adhesives was significantly reduced ($p < 0.05$), as shown in Fig. 6 (C).”

Table. 1 The specific p-values of Tukey's multiple comparisons in water solubility test.

	SB2	SPB	SBU
PU1	$P < 0.0001$	$P < 0.0001$	$P < 0.0001$
PU2	$P < 0.0001$	$P < 0.0001$	$P < 0.0001$
PU3	$P < 0.0001$	$P < 0.0001$	$P < 0.0001$
PU4	$P < 0.0001$	$P < 0.0001$	$P < 0.0001$
PU5	$P < 0.0001$	$P < 0.0001$	$P < 0.0001$
PU6	$P < 0.0001$	$P < 0.0001$	$P < 0.0001$
PU7	$P < 0.0001$	$P < 0.0001$	$P < 0.0001$

(2) “Fig. 6 (D) shows that the water absorption values of the seven PU adhesives are significantly lower than those of the three commercial adhesives ($p < 0.05$).”

Table. 2 The specific p-values of Tukey's multiple comparisons in water sorption test.

	SB2	SPB	SBU
PU1	$P < 0.0001$	$P = 0.0039$	$P < 0.0001$
PU2	$P < 0.0001$	$P = 0.0015$	$P < 0.0001$

PU3	P < 0.0001	P = 0.0009	P < 0.0001
PU4	P < 0.0001	P < 0.0001	P < 0.0001
PU5	P < 0.0001	P < 0.0001	P < 0.0001
PU6	P < 0.0001	P = 0.0013	P < 0.0001
PU7	P < 0.0001	P = 0.0003	P < 0.0001

(3) “The contact angles of the PU adhesives are all greater than 83°, significantly larger than that of the commercial adhesive ($p < 0.05$).”

Table. 3 The specific p-values of Tukey's multiple comparisons in contact angle test.

	SB2	SPB	SBU
PU1	P < 0.0001	P = 0.0081	P = 0.0042
PU2	P < 0.0001	P = 0.0003	P = 0.0001
PU3	P < 0.0001	P = 0.0007	P = 0.0003
PU4	P < 0.0001	P = 0.03	P = 0.0158
PU5	P < 0.0001	P = 0.0033	P = 0.0013
PU6	P < 0.0001	P = 0.0001	P < 0.0001
PU7	P < 0.0001	P = 0.0428	P = 0.0254

(4) “Fig. 6 (A) showed that there was no significant difference among PU2, PU3, and PU4, but their tensile strength was significantly higher than that of other groups ($p < 0.05$).”

Table. 4 The specific p-values of Tukey's multiple comparisons in tensile strength test.

	PU1	PU5	PU6	PU7
PU2	/	P = 0.0011	P < 0.0001	P < 0.0001
PU3	/	P = 0.0003	P < 0.0001	P < 0.0001
PU4	P = 0.0274	P < 0.0001	P < 0.0001	P < 0.0001

(5) “In terms of elongation at break, as shown in Fig. 6 (B), there was no significant difference between PU4 and PU5, and their elongation was obviously higher than that of the other groups ($p < 0.05$).”

Table. 5 The specific p-values of Tukey's multiple comparisons in elongation at break test.

	PU1	PU2	PU3	PU6	PU7
PU4	P < 0.0001	P < 0.0001	P < 0.0001	/	P < 0.0001
PU5	P < 0.0001	P < 0.0001	P < 0.0001	P = 0.0056	P < 0.0001

Letters indicating statistical difference in alphabetical order relating lower to higher means are preferable.

Letters indicating statistical difference in alphabetical order have been adjusted. And figures have been changed correspondingly.

Was microleakage qualitatively evaluated?

Microleakage was qualitatively evaluated as following. The microleakage of group SB2, group SPB and group SBU reached the axial surface with depth of 1.75 ± 0.08 mm, 1.79 ± 0.10 mm, and 1.70 ± 0.12 mm respectively. The leakage of group PU4 exceeded half of the depth of the hole, reaching 0.98 ± 0.16 mm. There is no significant difference among group SB2, group SPB and group SBU. However, the microleakage of commercial adhesive is significant higher than that of PU4 adhesive ($P < 0.0001$).

5) Z350XT is a conventional resin composite used as oblique and incremental technique. However, the author used this material as bulk fill, with 4-5 increment layer. This technique directly influences the adhesive performance, compromising the results and discussion.

I suggest that it be redone correctly (without research bias) or removed.

We are grateful for the suggestion.

In the manuscript, we described the using method of Z350XT as follows “The commercial composite resin Z350XT was placed on the surface of the treated dentin layer by layer approximately 4-5 mm in height with light curing for 40 s.” The describing phrase “layer by layer” means layered filling technique. We are sorry about the confusing writing, so we have modified the

manuscript describing this part to make it more clearly. The revised text is as following. “Three 1.5-mm-thick layers of commercial composite resin Z350XT were placed over the surface of the treated dentin. Each resin composite was light cured for 40 s using a light-curing unit.”

6) The thickness of resin composite directly influences on behavior of composite material, as well as the interface bonding. Therefore, it can be observed that fig. 9C presents a greater thickness compared to other images, influencing marginal microleakage in the back wall of the cavity (bias). Clarify.

We are grateful for the suggestion.

We are sorry about using the misleading picture. Now, fig. 9C was replaced with a new picture as following. The thickness of resin composite in four different groups was almost the same to ensure the reliability of the experiment.

Reviewer: 2

Comments:

1) Table 1 should be simplified. For example, the manufacturer's name could be removed, and the batch number could go as a table foot.

We are grateful for the suggestion.

Table 1 has already been simplified to make it more clearly. The manufacturer's name was removed, and the batch number went as a table foot.

Table 1. Commercial adhesive for this study

Material	Code	Category	Formulation
Single bond 2	SB2	2-step etch- and-rinse	bis-GMA, HEMA, dimethacrylates, silica nanofiller, polyalkenoic acid copolymer, initiators, water, ethanol
Spectrum bond	SPB	2-step etch- and-rinse	UDMA, trimethacrylate, PENTA, highly dispersed silicon dioxide, camphorquinone, BHT, cetylamine hydrofluoride, acetone
Single bond universal	SBU	Universal adhesive	MDP phosphate monomer, bis-GMA, dimethacrylate resins, HEMA, Vitrebond copolymer, fillers, ethanol, water, initiators, silane

* bis-GMA: bisphenol a diglycidyl methacrylate, HEMA: 2-hydroxyethyl methacrylate, UDMA: urethane dimethacrylate, PENTA: phosphoric acid modified acrylate resin, BHT: butylhydroxytoluene, MDP: methacryloyloxydecyl dihydrogenphosphate. Lot number: SB2 (N912223); SPB (1801000919); SBU (4330297).

2) Figure 6A and 6B show the mechanical properties of the synthesized PUs. However, these properties of commercial PUs are not shown. These data are also not included in the discussion of the results.

We are grateful for the suggestion.

The tensile strength and elongation at break is mainly used for testing materials with elasticity, such as the synthesized PU adhesive in this study. The composition of the commercial adhesive

system is different from that of the experimental PU adhesive. The commercial adhesive is mainly composed of the organic resin matrix and inorganic filler which is a rigid structure with almost no tensile deformation. Therefore, commercial adhesives were not tested. In this experiment, we make use of the tensile strength and elongation at break test to evaluate seven kinds of synthetic elastic PU adhesives. And combined with other tests (water absorption/solubility and contact angle), PU4 was considered to possess the best performance. And PU4 was chosen as the final experimental group to compare with three commercial adhesives in the follow-up tests.

3) Why are the conversion data for samples PU1, PU2, PU3, PU5 and PU6 not included?

We are grateful for the suggestion.

In this study, we prepared seven kinds of PU adhesives. Then, we conduct several experiments (tensile strength, elongation at break, water solubility, water sorption and contact angle) to evaluate them. Finally, among these seven adhesives, PU4 possesses the highest tensile strength and relatively higher elongation at break. Considering the water absorption/solubility and contact angle of seven kinds of PU adhesives comprehensively, PU4 was considered to have the best performance and chosen as the final experimental adhesive formulation for the following tests (degree of conversion, microtensile bond strength, microleakage and biocompatibility) to compare with three commercial adhesives. So, the conversion data for samples PU1, PU2, PU3, PU5 and PU6 was not included.

4) The microtensile bond strength can be related to tensile strength?. Explain it. By other hand, both microtensile and tensile are measured in the same scale (MPa)?

We are grateful for the suggestion.

Tensile strength is the maximum stress that a material can withstand while being stretched or pulled before breaking. In brittle materials the tensile strength is close to the yield point, whereas in ductile materials the tensile strength can be higher. The tensile strength is usually found by performing a tensile test. In this study, the dumbbell-shaped specimens were prepared in accordance with the standard ASTM-D638-2003 to evaluate the elastic property of PU adhesives. The prepared specimen is shown in Fig. 5 (H).

Microtensile bond strength test is currently considered as a versatile and standard bond strength testing method. It was introduced in 1994. Since then, it has been utilized profoundly across many bond strength testing laboratories, making it currently one of the most standard and versatile bond strength test.

In general, tensile strength is the elastic property of the material itself, however, microtensile bond strength is the adhesion ability of adhesive with tooth. There is no direct relationship between them.

And both microtensile bond strength and tensile strength are measured in the same scale (MPa). Microtensile bond strength is determined by dividing the loading force at break by the cross-sectional area of the sticks. Tensile strength is calculated by dividing the maximum stress at fracture by the cross-sectional area of the sample at the breaking point. So, they are measured in the same scale (MPa).

5) In the TGA curve there is a small inflection in the slope, if the curve was derived, it could detect PUA and PUB ?. Could you show DSC of PUA, PUB and PU4?.

Thank you very much for your comments. We are grateful for the suggestion. Your comments are very valuable for us to improve the thermal stability of the PU adhesive material. The TGA curve is the result of final formulation of PU4. TGA of separate PUA and PUB need to be tested. Due to COVID-19, the school has been closed. We contacted the school laboratory and it was uncertainty when it would be available. Therefore, the supplement results of TGA and DSC may need waiting a long time. The TGA of PU4 tested in this study is not enough for evaluating the thermal stability of the material which is a limitation of the study, but it can also show some information about the PU4 adhesive. The initial degradation temperature of 5% weight loss was observed at 269.00 °C. The maximum tolerant temperature of the oral mucosa is approximately 60 °C. Therefore, PU4 adhesive may be applied in the oral environment. We searched the relevant literature, and the following two articles can be used for reference. Our laboratory will continue to

study in this area. In the subsequent research, we will conduct more detailed and complete experiments of the thermal stability of the material.

1. Solís-Correa R, Vargas-Coronado R, Aguilar-Vega M, Cauch-Rodríguez J, San Román J, Marco A. Synthesis of HMDI-based segmented polyurethanes and their use in the manufacture of elastomeric composites for cardiovascular applications. *J Biomater Sci Polym Ed.* 2007;18(5):561-78.
2. Gong H, Guo X, Cao D, Gao P, Feng D, Zhang X, Shi Z, Zhang Y, Zhu S, Cui Z. Photopolymerizable and moisture-curable polyurethanes for dental adhesive applications to increase restoration durability. *J Mater Chem B.* 2019;7(5):744-54.

6) Figure 8F shows that SBU aging can also penetrate the dentinal tubules, and Figure 6G shows that the microtensile bond strength of SBU is greater than PU4 ?. Could you say that by this measure that PU4 is not better than SBU?.

We are grateful for the suggestion.

Although SBU aging can also penetrate the dentinal tubules, its adhesive layer fractured more severely. In the aging group, microtensile bond strength of SBU is greater than that of PU4, but there is no statistical significance. The microtensile bond strength of SBU decreased obviously after aging, while the microtensile bond strength of PU4 increased. It is indicated that longer-term aging test may be needed in the future study. In term of the microtensile experiment, SBU is better than PU4, but water absorption and water solubility of PU4 are much lower than that of SBU which may contribute to reducing hydrolytic degradation of the bonding interface. And microleakage of PU4 has also been improved when compared with SBU. Its elastic properties can buffer various stresses during long-term use, improving the stability and durability of the adhesive bonding interface. Therefore, the comprehensive performance of PU4 is still better than that of SBU.

7) In Figure 10A, the viability values are not shown in percentages as indicated (i.e., 1.0= 100%). Setting viability at 70% is arbitrary (the reader can suppose a 30% death of fibroblasts in the mouth?). What is the variation in your controls?. Even a 20% decrease in viability, this is significant in a cell population?. You can perform an ANOVA followed by a Tukey test to set significance to 5%.

We are grateful for the suggestion.

In Figure 10A, the viability values have been shown in percentages in the manuscript.

According to GB/T 16886.5-2003 (ISO 10993-5:1999), samples with cell viability larger than 75% of blank group can be considered as non-cytotoxicity[1-3]. We are sorry that 70% is not accurate. It has been revised in the text. In this study, cell viability of PU4 adhesive was all greater than 80% which demonstrated good cell viability and biocompatibility, meeting the clinical biosafety requirements.

The variation of controls has already been added.

We have performed an ANOVA followed by a Tukey test. In the group of 48 h and 72 h, there is significant difference between control group and the other groups (SB2, SPB, SBU, PU4). However, no statistical difference exists between PU4 adhesive and three commercial adhesives.

It is important to mention that the commercial adhesive system may show some toxicity to a certain extent. Adhesive systems components [e.g. Bis-GMA (bisphenol A diglycidyl methacrylate), HEMA (2-hydroxyethyl methacrylate), TEGDMA (tri-ethylene glycol dimethacrylate), and UDMA (urethane dimethacrylate), camphorquinone] are cytotoxic. Despite its cytotoxic produced by direct contact on pulp cells, these may be clinically attenuated or neutralized, as adhesive systems are applied to dentin, which acts as a physical barrier depending on its thickness.

In this study, there is no significant difference between commercial adhesives which is used in clinical practice and PU adhesives in MTT test which indicates that PU4 adhesive can meet the clinical biosafety requirements.

1. Cao D, Zhang Y, Li Y, Shi X, Gong H, Feng D, Guo X, Shi Z, Zhu S, Cui Z. Fabrication of superhydrophobic coating for preventing microleakage in a dental composite restoration. *Mater Sci Eng C Mater Biol Appl.* 2017;78:333-40.
2. Yang X, Yang K, Yu F, Chen X, Wu S, Zhu Z. Preparation of novel bilayer hydrogels by combination of irradiation and freeze-thawing and their physical and biological properties. *Polym Int.* 2009; 58:1291-8.

3. Sudarsan S, Franklin D S, Sakthivel M, Guhanathan S. Non toxic, antibacterial, biodegradable hydrogels with pH-stimuli sensitivity: Investigation of swelling parameters. Carbohydr Polym. 2016; 148:206-15.

8) The light microscopy images in Figure 10B must be improved, or show SEM images, in order to visualize the morphology of the cells. With the viability shown in Figure 10A, some dead (red) cells should be observed. Fibroblasts grow extended (in spreading), however the fluorescence image gives the appearance of cluster or aggregates. Can you explain it?

We are grateful for the suggestion.

The light microscopy images in Figure 10B have been improved to make it more clearly. And we adjusted the contrast of the fluorescence images, and the dead cells (red) could be seen more clearly.

L929 fibroblasts should grow extended (in spreading), however the fluorescence image gives the appearance of cluster or aggregates. In this study, the live/dead cell staining kit includes two

parts that calcein-AM is used for living cell staining and propidium iodide is used for dead cell staining. The staining solution is slightly toxic to the cells. If the staining time is too long, it may cause poor cell morphology. And the osmotic pressure of the staining solution may not be exactly the same with the cytoplasm which can also lead to the cell deformation.

Appendix B**ROYAL SOCIETY
OPEN SCIENCE****Hydrolysis-resistant and stress-buffering
bifunctional polyurethane adhesive for durable dental
composite restoration**

Journal:	Royal Society Open Science
Manuscript ID	RSOS-200457.R1
Article Type:	Research
Date Submitted by the Author:	14-May-2020
Complete List of Authors:	zhang, Jiahui; Jilin University, Guo, Xiaowei; Jilin University Zhang, Xiaomeng; Jilin University Wang, Huimin; Jilin University Zhu, Jiufu; Jilin University Shi, Zuosen; Jilin University Zhu, Song; Jilin University Cui, Zhanchen; Jilin University
Subject:	Materials science < CHEMISTRY, biomaterials < CROSS-DISCIPLINARY SCIENCES
Keywords:	polyurethane adhesive, dental restoration, hydrolysis-resistant, stress-buffering
Subject Category:	Chemistry

Author-supplied statements

Relevant information will appear here if provided.

Ethics

Does your article include research that required ethical approval or permits?:

Yes

Statement (if applicable):

All relevant ethical applications (permit no. KT202003081) in this experiment were approved by the Ethics Committee of School and Hospital of Stomatology of Jilin University. Informed consent was obtained from all donors.

Data

It is a condition of publication that data, code and materials supporting your paper are made publicly available. Does your paper present new data?:

Yes

Statement (if applicable):

The data is provided as electronic supplementary materials.

Conflict of interest

I/We declare we have no competing interests

Statement (if applicable):

CUST_STATE_CONFLICT :No data available.

Authors' contributions

This paper has multiple authors and our individual contributions were as below

Statement (if applicable):

Authors' major contribution:

Jiahui Zhang: conducted the statistical analysis and wrote this manuscript;

Xiaowei Guo and Xiaomeng Zhang: finish the acquisition of data;

Huimin Wang, Jiufu Zhu and Zuosen Shi: conduct the experiments and repeated them;

Corresponding author: Song Zhu and Zhanchen Cui: designed and oversaw the study, and reviewed and revised this manuscript.

Point by point response to the reviewers' comments

Dear Editor and Reviewers,

We hope you are keeping well at this difficult time.

Thanks very much for taking your time to review this manuscript. I really appreciate all these precious comments and suggestions. Please find my itemized responses in below and my revisions in the resubmitted files.

Thanks again.

Reviewer: 1

1) Abstract.

It's necessary that all methodologies performed are specified.

The conclusion does not answer the objective of the paper.

We are grateful for the suggestion.

We have revised the abstract according to the suggestion that all methodologies performed are specified and the conclusion answers the objective of the paper. The revised text is as following. "A new elastic polyurethane (PU) adhesive was reported in this study to improve the stability and durability of the dental adhesion interface. A polyurethane oligomer was synthesized by the solution polymerization method, and a diluent and solvent were added to prepare PU adhesives. The water sorption, water solubility, contact angle, thermal stability, **degree of conversion** and mechanical properties of the PU adhesives were evaluated. Experimental applications for tooth restoration (**microtensile bond strength and microleakage**) were also performed. **And cytotoxicity test was carried out.** The water sorption and solubility of the PU adhesives were significantly lower than those of three commercial adhesives. The microtensile bond strength of the PU adhesives was improved after thermocycling test, and the extent of microleakage was diminished when compared with that of commercial adhesives. Biocompatibility testing demonstrated that the PU adhesive was nontoxic to L929 fibroblasts. This study **shows the ability of PU adhesive to improve the stability and durability of the dental adhesion interface** and may refocus the attention of scientists from rigid bonding to flexible bonding for dental adhesion, and it sheds light on a new strategy for the stable and durable bonding interface of dentin adhesives."

2) Fig.1 and 3 are unnecessary. Graphic abstract is more suitable.

We are grateful for the suggestion.

Based on the comments, Fig.1 and 3 were removed in the article and graphic abstract was added.

3) Some methodologies used are insufficient described, as contact angle (how the calculation was performed; kind of liquid; device information; software used); thermal stability characterization; failure pattern in SEM.

We are grateful for the suggestion.

More detailed information was added to supplement the insufficient methodologies according to the comment to make it more clearly. The revised text is as following.

contact angle

“Contact angles were obtained using the sessile drop method with a Dataphysics contact angle analyzer (OCA-20, DataPhysics Co., German). This instrument consists of a CCD video camera with a resolution of 768×576 pixel and up to 50 images per second and multiple microsyringe units. A drop of $6 \mu\text{l}$ of deionized water was gently dropped onto the surface of the adhesive to take a digital photo. The digital drop image was processed by a specialized software SCA 20, which calculated both the left and right contact angles from the shape of the drop.” The representative digital drop images are shown in the following picture.

thermal stability characterization;

“Thermogravimetric analysis measurement of PU4 adhesive was performed using a TGA thermal analyzer (Q500, TA Instruments, USA). Initial sample masse is around 5 mg. The heating rate is $10 \text{ }^\circ\text{C}/\text{min}$. The experiments were performed in an inert atmosphere with a continuous flow of nitrogen at the rate of $150 \text{ ml}/\text{min}$ and heated from room temperature to $800 \text{ }^\circ\text{C}$.”

failure pattern in SEM.

The SEM in this study is the intact bonding surface of the PU4 adhesive and three commercial
adhesives. We are sorry that failure pattern in SEM was not conducted in this study. In the following
study, we will supplement this experiment according to the suggestion.

Which light-curing unit was used?

The light-curing unit was a LED light (SLC-VIIIA, China) which was used in clinical practice.
The LED light had an output light intensity of approximately 900 mW/cm², which was monitored
by a radiometer to ensure light intensity.

Only PU4 was used as experimental group for several methodologies. Clarify.

In this study, we prepared seven kinds of PU adhesives. Then, we conduct several experiments
(tensile strength, elongation at break, water solubility, water sorption and contact angle) to evaluate
them. Finally, among these seven adhesives, PU4 possesses the highest tensile strength and
relatively higher elongation at break. Considering the water absorption/solubility and contact angle
of seven kinds of PU adhesives comprehensively, PU4 was considered to have the best
comprehensive performance and chosen as the final experimental adhesive formulation for the
following tests (degree of conversion, microtensile bond strength, microleakage and
biocompatibility) to compare with three commercial adhesives. So, only PU4 was used as
experimental group for the following methodologies.

4) Statistical test of normality and homoscedasticity distribution is required. Only two-way
ANOVA is presented for Microtensile bond strength with uncertainty post-hoc test (Tukey or
Dunnet?). It is necessary to present all statistical analyzes for all the methodologies developed.

We are grateful for the suggestion.

Statistical test of normality and homoscedasticity distribution has been conducted. All statistical
analyzes for all the methodologies developed have been presented in the section of statistical
analysis. We have added this section in the text of statistical analysis to make it more clearly. The
revised text is as following.

“Data were expressed as the mean \pm standard deviation. The data were consistent with
normality and homoscedasticity distribution. Data for microtensile bond strength were analyzed

using two-way ANOVA, and the data of tensile strength, elongation at break, water solubility, water sorption, contact angle and degree of conversion were submitted to one-way ANOVA using SPSS software (version 19.0, SPSS Inc., Chicago, IL, USA). Multiple comparison analysis was conducted using the Tukey test. The significance level was set at $p = 0.05$ for this study.”

In the results topic, p-values are presented ($p < 0.05$). It is required to present the specific p-value, when p is statistically significant.

The specific p-values have been presented when p is statistically significant in the manuscript. However, there are several summary statements with more than one P value. Therefore, we have listed them in the following tables to ensure the conciseness of the article.

(1) “Compared with these methacrylic resin adhesives, the solubility of PU adhesives was significantly reduced ($p < 0.05$), as shown in Fig. 6 (C).”

Table. 1 The specific p-values of Tukey's multiple comparisons in water solubility test.

	SB2	SPB	SBU
PU1	$P < 0.0001$	$P < 0.0001$	$P < 0.0001$
PU2	$P < 0.0001$	$P < 0.0001$	$P < 0.0001$
PU3	$P < 0.0001$	$P < 0.0001$	$P < 0.0001$
PU4	$P < 0.0001$	$P < 0.0001$	$P < 0.0001$
PU5	$P < 0.0001$	$P < 0.0001$	$P < 0.0001$
PU6	$P < 0.0001$	$P < 0.0001$	$P < 0.0001$
PU7	$P < 0.0001$	$P < 0.0001$	$P < 0.0001$

(2) “Fig. 6 (D) shows that the water absorption values of the seven PU adhesives are significantly lower than those of the three commercial adhesives ($p < 0.05$).”

Table. 2 The specific p-values of Tukey's multiple comparisons in water sorption test.

	SB2	SPB	SBU
PU1	$P < 0.0001$	$P = 0.0039$	$P < 0.0001$
PU2	$P < 0.0001$	$P = 0.0015$	$P < 0.0001$

PU3	P < 0.0001	P = 0.0009	P < 0.0001
PU4	P < 0.0001	P < 0.0001	P < 0.0001
PU5	P < 0.0001	P < 0.0001	P < 0.0001
PU6	P < 0.0001	P = 0.0013	P < 0.0001
PU7	P < 0.0001	P = 0.0003	P < 0.0001

(3) “The contact angles of the PU adhesives are all greater than 83°, significantly larger than that of the commercial adhesive ($p < 0.05$).”

Table. 3 The specific p-values of Tukey's multiple comparisons in contact angle test.

	SB2	SPB	SBU
PU1	P < 0.0001	P = 0.0081	P = 0.0042
PU2	P < 0.0001	P = 0.0003	P = 0.0001
PU3	P < 0.0001	P = 0.0007	P = 0.0003
PU4	P < 0.0001	P = 0.03	P = 0.0158
PU5	P < 0.0001	P = 0.0033	P = 0.0013
PU6	P < 0.0001	P = 0.0001	P < 0.0001
PU7	P < 0.0001	P = 0.0428	P = 0.0254

(4) “Fig. 6 (A) showed that there was no significant difference among PU2, PU3, and PU4, but their tensile strength was significantly higher than that of other groups ($p < 0.05$).”

Table. 4 The specific p-values of Tukey's multiple comparisons in tensile strength test.

	PU1	PU5	PU6	PU7
PU2	/	P = 0.0011	P < 0.0001	P < 0.0001
PU3	/	P = 0.0003	P < 0.0001	P < 0.0001
PU4	P = 0.0274	P < 0.0001	P < 0.0001	P < 0.0001

(5) “In terms of elongation at break, as shown in Fig. 6 (B), there was no significant difference between PU4 and PU5, and their elongation was obviously higher than that of the other groups ($p < 0.05$).”

Table. 5 The specific p-values of Tukey's multiple comparisons in elongation at break test.

	PU1	PU2	PU3	PU6	PU7
PU4	$P < 0.0001$	$P < 0.0001$	$P < 0.0001$	/	$P < 0.0001$
PU5	$P < 0.0001$	$P < 0.0001$	$P < 0.0001$	$P = 0.0056$	$P < 0.0001$

Letters indicating statistical difference in alphabetical order relating lower to higher means are preferable.

Letters indicating statistical difference in alphabetical order have been adjusted. And figures have been changed correspondingly.

Was microleakage qualitatively evaluated?

Microleakage was qualitatively evaluated as following. The microleakage of group SB2, group SPB and group SBU reached the axial surface with depth of 1.75 ± 0.08 mm, 1.79 ± 0.10 mm, and 1.70 ± 0.12 mm respectively. The leakage of group PU4 exceeded half of the depth of the hole, reaching 0.98 ± 0.16 mm. There is no significant difference among group SB2, group SPB and group SBU. However, the microleakage of commercial adhesive is significant higher than that of PU4 adhesive ($P < 0.0001$).

5) Z350XT is a conventional resin composite used as oblique and incremental technique. However, the author used this material as bulk fill, with 4-5 increment layer. This technique directly influences the adhesive performance, compromising the results and discussion.

I suggest that it be redone correctly (without research bias) or removed.

We are grateful for the suggestion.

In the manuscript, we described the using method of Z350XT as follows “The commercial composite resin Z350XT was placed on the surface of the treated dentin layer by layer approximately 4-5 mm in height with light curing for 40 s.” The describing phrase “layer by layer” means layered filling technique. We are sorry about the confusing writing, so we have modified the

manuscript describing this part to make it more clearly. The revised text is as following. “Three 1.5-
5 mm-thick layers of commercial composite resin Z350XT were placed over the surface of the treated
dentin. Each resin composite was light cured for 40 s using a light-curing unit.”

6) The thickness of resin composite directly influences on behavior of composite material, as well
as the interface bonding. Therefore, it can be observed that fig. 9C presents a greater thickness
compared to other images, influencing marginal microleakage in the back wall of the cavity (bias).
Clarify.

We are grateful for the suggestion.

We are sorry about using the misleading picture. Now, fig. 9C was replaced with a new picture
as following. The thickness of resin composite in four different groups was almost the same to
ensure the reliability of the experiment.

Reviewer: 2

Comments:

1) Table 1 should be simplified. For example, the manufacturer's name could be removed, and the batch number could go as a table foot.

We are grateful for the suggestion.

Table 1 has already been simplified to make it more clearly. The manufacturer's name was removed, and the batch number went as a table foot.

Table 1. Commercial adhesive for this study

Material	Code	Category	Formulation
Single bond 2	SB2	2-step etch- and-rinse	bis-GMA, HEMA, dimethacrylates, silica nanofiller, polyalkenoic acid copolymer, initiators, water, ethanol
Spectrum bond	SPB	2-step etch- and-rinse	UDMA, trimethacrylate, PENTA, highly dispersed silicon dioxide, camphorquinone, BHT, cetylamine hydrofluoride, acetone
Single bond universal	SBU	Universal adhesive	MDP phosphate monomer, bis-GMA, dimethacrylate resins, HEMA, Vitrebond copolymer, fillers, ethanol, water, initiators, silane

* bis-GMA: bisphenol a diglycidyl methacrylate, HEMA: 2-hydroxyethyl methacrylate, UDMA: urethane dimethacrylate, PENTA: phosphoric acid modified acrylate resin, BHT: butylhydroxytoluene, MDP: methacryloyloxydecyl dihydrogenphosphate. Lot number: SB2 (N912223); SPB (1801000919); SBU (4330297).

2) Figure 6A and 6B show the mechanical properties of the synthesized PUs. However, these properties of commercial PUs are not shown. These data are also not included in the discussion of the results.

We are grateful for the suggestion.

The tensile strength and elongation at break is mainly used for testing materials with elasticity, such as the synthesized PU adhesive in this study. The composition of the commercial adhesive

system is different from that of the experimental PU adhesive. The commercial adhesive is mainly
composed of the organic resin matrix and inorganic filler which is a rigid structure with almost no
tensile deformation. Therefore, commercial adhesives were not tested. In this experiment, we make
use of the tensile strength and elongation at break test to evaluate seven kinds of synthetic elastic
PU adhesives. And combined with other tests (water absorption/solubility and contact angle), PU4
was considered to possess the best performance. And PU4 was chosen as the final experimental
group to compare with three commercial adhesives in the follow-up tests.

3) Why are the conversion data for samples PU1, PU2, PU3, PU5 and PU6 not included?

We are grateful for the suggestion.

In this study, we prepared seven kinds of PU adhesives. Then, we conduct several experiments
(tensile strength, elongation at break, water solubility, water sorption and contact angle) to evaluate
them. Finally, among these seven adhesives, PU4 possesses the highest tensile strength and
relatively higher elongation at break. Considering the water absorption/solubility and contact angle
of seven kinds of PU adhesives comprehensively, PU4 was considered to have the best performance
and chosen as the final experimental adhesive formulation for the following tests (degree of
conversion, microtensile bond strength, microleakage and biocompatibility) to compare with three
commercial adhesives. So, the conversion data for samples PU1, PU2, PU3, PU5 and PU6 was not
included.

4) The microtensile bond strength can be related to tensile strength?. Explain it. By other hand,
both microtensile and tensile are measured in the same scale (MPa)?

We are grateful for the suggestion.

Tensile strength is the maximum stress that a material can withstand while being stretched or
pulled before breaking. In brittle materials the tensile strength is close to the yield point, whereas in
ductile materials the tensile strength can be higher. The tensile strength is usually found by
performing a tensile test. In this study, the dumbbell-shaped specimens were prepared in accordance
with the standard ASTM-D638-2003 to evaluate the elastic property of PU adhesives. The prepared
specimen is shown in Fig. 5 (H).

Microtensile bond strength test is currently considered as a versatile and standard bond strength testing method. It was introduced in 1994. Since then, it has been utilized profoundly across many bond strength testing laboratories, making it currently one of the most standard and versatile bond strength test.

In general, tensile strength is the elastic property of the material itself, however, microtensile bond strength is the adhesion ability of adhesive with tooth. There is no direct relationship between them.

And both microtensile bond strength and tensile strength are measured in the same scale (MPa). Microtensile bond strength is determined by dividing the loading force at break by the cross-sectional area of the sticks. Tensile strength is calculated by dividing the maximum stress at fracture by the cross-sectional area of the sample at the breaking point. So, they are measured in the same scale (MPa).

5) In the TGA curve there is a small inflection in the slope, if the curve was derived, it could detect PUA and PUB ?. Could you show DSC of PUA, PUB and PU4?.

Thank you very much for your comments. We are grateful for the suggestion. Your comments are very valuable for us to improve the thermal stability of the PU adhesive material. The TGA curve is the result of final formulation of PU4. TGA of separate PUA and PUB need to be tested. Due to COVID-19, the school has been closed. We contacted the school laboratory and it was uncertainty when it would be available. Therefore, the supplement results of TGA and DSC may need waiting a long time. The TGA of PU4 tested in this study is not enough for evaluating the thermal stability of the material which is a limitation of the study, but it can also show some information about the PU4 adhesive. The initial degradation temperature of 5% weight loss was observed at 269.00 °C. The maximum tolerant temperature of the oral mucosa is approximately 60 °C. Therefore, PU4 adhesive may be applied in the oral environment. We searched the relevant literature, and the following two articles can be used for reference. Our laboratory will continue to

study in this area. In the subsequent research, we will conduct more detailed and complete
experiments of the thermal stability of the material.

1. Solís-Correa R, Vargas-Coronado R, Aguilar-Vega M, Cauich-Rodríguez J, San Román J, Marco A. Synthesis of HMDI-based segmented polyurethanes and their use in the manufacture of elastomeric composites for cardiovascular applications. *J Biomater Sci Polym Ed.* 2007;18(5):561-78.
 2. Gong H, Guo X, Cao D, Gao P, Feng D, Zhang X, Shi Z, Zhang Y, Zhu S, Cui Z. Photopolymerizable and moisture-curable polyurethanes for dental adhesive applications to increase restoration durability. *J Mater Chem B.* 2019;7(5):744-54.

6) Figure 8F shows that SBU aging can also penetrate the dentinal tubules, and Figure 6G shows that the microtensile bond strength of SBU is greater than PU4 ?. Could you say that by this measure that PU4 is not better than SBU?.

We are grateful for the suggestion.

Although SBU aging can also penetrate the dentinal tubules, its adhesive layer fractured more severely. In the aging group, microtensile bond strength of SBU is greater than that of PU4, but there is no statistical significance. The microtensile bond strength of SBU decreased obviously after aging, while the microtensile bond strength of PU4 increased. It is indicated that longer-term aging test may be needed in the future study. In term of the microtensile experiment, SBU is better than PU4, but water absorption and water solubility of PU4 are much lower than that of SBU which may contribute to reducing hydrolytic degradation of the bonding interface. And microleakage of PU4 has also been improved when compared with SBU. Its elastic properties can buffer various stresses during long-term use, improving the stability and durability of the adhesive bonding interface. Therefore, the comprehensive performance of PU4 is still better than that of SBU.

7) In Figure 10A, the viability values are not shown in percentages as indicated (i.e., 1.0= 100%). Setting viability at 70% is arbitrary (the reader can suppose a 30% death of fibroblasts in the mouth?). What is the variation in your controls?. Even a 20% decrease in viability, this is significant in a cell population?. You can perform an ANOVA followed by a Tukey test to set significance to 5%.

We are grateful for the suggestion.

In Figure 10A, the viability values have been shown in percentages in the manuscript.

According to GB/T 16886.5-2003 (ISO 10993-5:1999), samples with cell viability larger than 75% of blank group can be considered as non-cytotoxicity[1-3]. We are sorry that 70% is not accurate. It has been revised in the text. In this study, cell viability of PU4 adhesive was all greater than 80% which demonstrated good cell viability and biocompatibility, meeting the clinical biosafety requirements.

The variation of controls has already been added.

We have performed an ANOVA followed by a Tukey test. In the group of 48 h and 72 h, there is significant difference between control group and the other groups (SB2, SPB, SBU, PU4). However, no statistical difference exists between PU4 adhesive and three commercial adhesives.

It is important to mention that the commercial adhesive system may show some toxicity to a certain extent. Adhesive systems components [e.g. Bis-GMA (bisphenol A diglycidyl methacrylate), HEMA (2-hydroxyethyl methacrylate), TEGDMA (tri-ethylene glycol dimethacrylate), and UDMA (urethane dimethacrylate), camphorquinone] are cytotoxic. Despite its cytotoxic produced by direct contact on pulp cells, these may be clinically attenuated or neutralized, as adhesive systems are applied to dentin, which acts as a physical barrier depending on its thickness.

In this study, there is no significant difference between commercial adhesives which is used in clinical practice and PU adhesives in MTT test which indicates that PU4 adhesive can meet the clinical biosafety requirements.

1. Cao D, Zhang Y, Li Y, Shi X, Gong H, Feng D, Guo X, Shi Z, Zhu S, Cui Z. Fabrication of superhydrophobic coating for preventing microleakage in a dental composite restoration. *Mater Sci Eng C Mater Biol Appl.* 2017;78:333-40.
2. Yang X, Yang K, Yu F, Chen X, Wu S, Zhu Z. Preparation of novel bilayer hydrogels by combination of irradiation and freeze-thawing and their physical and biological properties. *Polym Int.* 2009; 58:1291-8.

3. Sudarsan S, Franklin D S, Sakthivel M, Guhanathan S. Non toxic, antibacterial, biodegradable
hydrogels with pH-stimuli sensitivity: Investigation of swelling parameters. Carbohydr Polym.
2016; 148:206-15.

8) The light microscopy images in Figure 10B must be improved, or show SEM images, in order to
visualize the morphology of the cells. With the viability shown in Figure 10A, some dead (red) cells
should be observed. Fibroblasts grow extended (in spreading), however the fluorescence image
gives the appearance of cluster or aggregates. Can you explain it?

We are grateful for the suggestion.

The light microscopy images in Figure 10B have been improved to make it more clearly. And
we adjusted the contrast of the fluorescence images, and the dead cells (red) could be seen more
clearly.

L929 fibroblasts should grow extended (in spreading), however the fluorescence image gives the appearance of cluster or aggregates. In this study, the live/dead cell staining kit includes two

parts that calcein-AM is used for living cell staining and propidium iodide is used for dead cell
staining. The staining solution is slightly toxic to the cells. If the staining time is too long, it may
cause poor cell morphology. And the osmotic pressure of the staining solution may not be exactly
the same with the cytoplasm which can also lead to the cell deformation.

Hydrolysis-resistant and stress-buffering bifunctional polyurethane adhesive for durable dental composite restoration

Jiahui Zhang^a, Xiaowei Guo^a, Xiaomeng Zhang^a, Huimin Wang^a, Jiufu Zhu^b, Zuosen Shi^b, Song Zhu^{a*}, Zhanchen Cui^{b*}

^aDepartment of Prosthetic Dentistry, Hospital of Stomatology, Jilin University, Changchun, 130012, PR China

^bState Key Lab of Supramolecular Structure and Materials, College of Chemistry, Jilin University, Changchun, 130012, PR China

Jiahui Zhang: 290088672@qq.com

Xiaowei Guo: 1539769099@qq.com

Xiaomeng Zhang: 408199196@qq.com

Huimin Wang: 2781258252@qq.com

Jiufu Zhu: 627050733@qq.com

Zuosen Shi: shizs@jlu.edu.cn

Corresponding author: Song Zhu and Zhanchen Cui

E-mail addresses: zhusong1965@163.com (S. Zhu), cuizc@jlu.edu.cn (Z. Cui).

Abstract

A new elastic polyurethane (PU) adhesive was reported in this study to improve the stability and durability of the dental adhesion interface. A polyurethane oligomer was synthesized by the solution polymerization method, and a diluent and solvent were added to prepare PU adhesives. The water sorption, water solubility, contact angle, thermal stability, **degree of conversion** and mechanical properties of the PU adhesives were evaluated. Experimental applications for tooth restoration (**microtensile bond strength and microleakage**) were also performed. **And cytotoxicity test was carried out.** The water sorption and solubility of the PU adhesives were significantly lower than those of three commercial adhesives. The microtensile bond strength of the PU adhesives was improved after thermocycling test, and the extent of microleakage was diminished when compared with that of commercial adhesives. Biocompatibility testing demonstrated that the PU adhesive was nontoxic to L929 fibroblasts. This study **shows the ability of PU adhesive to improve the stability and durability of the dental adhesion interface** and may refocus the attention of scientists from rigid bonding to flexible bonding for dental adhesion, and it sheds light on a new strategy for the stable and durable bonding interface of dentin adhesives.

Keywords: polyurethane adhesive; dental restoration; hydrolysis-resistant; stress-buffering

1 Introduction

Composite resin has been widely used in dental restoration for more than 60 years due to its aesthetic advantages, excellent mechanical properties, ease of use and acceptable price(1-4). The success of composite resin restoration relies on bonding techniques that can bond these plastic materials to the tissue of teeth. Therefore, strong and durable bonding properties are necessary for successful composite resin restoration(5-8).

The failure of restorations is mainly due to defects in the bonding interface, which are caused by the polymerization stress when the composite resin is polymerized using a curing light(9). Scientists have made many efforts to reduce the polymerization shrinkage of composite resins, for example, by using low-shrinkage composite resin. It has been reported that the volume of polymerization shrinkage of the composite resin can be reduced to less than 1%, and the generation of gaps between the tooth and the composite resin can be temporarily avoided(10, 11). However, there will be continuous

mechanical chewing stress in the mouth after dental restoration. Moreover, studies have
shown that the thermal expansion coefficient of the composite resin is usually $2.0 \times$
$10^{-3} \text{ \%/}^\circ\text{C}$, which is larger than that of dentin (approximately $1.1 \times 10^{-3} \text{ \%/}^\circ\text{
[revised manuscript text omitted]

Table 2. Formulation of 7 kinds of PU adhesives

	PUA	PUB	HEMA	TEGDMA	Acetone	CQ	4-EDMAB
PU1	12g	0g	1.71g	1.71g	1.54g	0.05g	0.12g
PU2	9g	3g	1.71g	1.71g	1.54g	0.05g	0.12g
PU3	8g	4g	1.71g	1.71g	1.54g	0.05g	0.12g
PU4	6g	6g	1.71g	1.71g	1.54g	0.05g	0.12g
PU5	4g	8g	1.71g	1.71g	1.54g	0.05g	0.12g
PU6	3g	9g	1.71g	1.71g	1.54g	0.05g	0.12g
PU7	0g	12g	1.71g	1.71g	1.54g	0.05g	0.12g

PU adhesive = 70% PU (PUA + PUB) + 10% HEMA + 10% TEGDMA + 9% acetone + 0.3% CQ + 0.7% 4-EDMAB

The structure of the polyurethane oligomer is characterized by Fourier transform infrared spectroscopy (FTIR) and nuclear magnetic resonance spectroscopy (^1H NMR spectrum). FTIR was measured by a BRUKER VERTEX 80 V infrared spectrometer in the range of $4000\text{-}500\text{ cm}^{-1}$. The ^1H NMR spectrum was measured by a Bruker AVANCE 500 MHz type III nuclear magnetic resonance spectrometer using deuterated chloroform as a solvent.

2.3 Water sorption(W_{SP}) and water solubility(W_{SL})

Water absorption and water solubility were determined according to ISO 4049:2009. Disk-shaped samples ($d = 15.0\text{ mm}$, $h = 1.0\text{ mm}$, $n = 5$) were prepared. The polyurethane adhesive was poured into the mold, covered with a piece of polyester film, and cured with a light intensity of 900 mW/cm^2 for 10 s. The curing light unit was monitored by a radiometer to ensure light intensity. All the samples prepared were placed in a desiccator with silica gel at $37 \pm 2\text{ }^\circ\text{C}$ for 24 h. The samples were then transferred into another desiccator for 2 h at $23 \pm 1\text{ }^\circ\text{C}$ and weighed. This process was repeated until a constant mass (M_1) was obtained. The diameter and thickness of each sample were measured by electronic digital caliper to calculate the volume (V ; mm^3) of the sample. Each sample was then immersed in a sealed glass vial containing 15 ml of deionized water and soaked for 7 days at $37 \pm 1\text{ }^\circ\text{C}$. The samples were rinsed with running deionized water, and the surface water was dried with filter paper. Then, the samples were weighed to obtain mass M_2 . The samples were redried in a $37 \pm 1\text{ }^\circ\text{C}$

desiccator, as described above, until a stable mass M3 was obtained. The calculation formula for water absorption and solubility of the sample is as follows:

$$W_{SP}=(M2-M3)/V$$

$$W_{SL}=(M1-M3)/V$$

2.4 Contact angle measurements

Contact angles were obtained using the sessile drop method with a Dataphysics contact angle analyzer (OCA-20, DataPhysics Co., German). This instrument consists of a CCD video camera with a resolution of 768×576 pixel and up to 50 images per second and multiple microsyringe units. A drop of 6 μ l of deionized water was gently dropped onto the surface of the adhesive to take a digital photo. The digital drop image was processed by a specialized software SCA 20, which calculated both the left and right contact angles from the shape of the drop.

2.5 Tensile strength and elongation at break of PU adhesives

The dumbbell-shaped specimens (n=5) were prepared in accordance with the standard ASTM-D638-2003. The prepared specimen is shown in Fig. 5 (H). The specimen was tested using a universal testing machine (AG-X plus, Shimadzu Corporation, Japan) with a crosshead speed of 10 mm/min until it was broken.

2.6 Thermal stability characterization

Thermogravimetric analysis measurement of PU4 adhesive was performed using a TGA thermal analyzer (Q500, TA Instruments, USA). Initial sample masse is around 5 mg. The heating rate is 10 $^{\circ}$ C/min. The experiments were performed in an inert atmosphere with a continuous flow of nitrogen at the rate of 150 ml/min and heated from room temperature to 800 $^{\circ}$ C.

2.7 Degree of conversion

The degree of conversion (DC) of PU4 adhesives and three commercial adhesives were measured. The DC was determined by a Fourier transform infrared spectrometer equipped with an attenuated total reflectance device for 5 samples per group (n=5). The FTIR of uncured adhesive was obtained as a control. The adhesive was cured for 10 s, and the polymerized adhesive was immediately subjected to FTIR. After light-curing,

the area of infrared absorption peak of methacrylate double bonds (C = C, peak at 1637
5 cm⁻¹) decreased, and the carbonyl group (peak at 1720 cm⁻¹) was used as the internal
standard. The calculation of the DC used the following equation:

$$8 \quad DC\% = \left[1 - \frac{(A_{1636}/A_{1720})_{\text{peak area after curing}}}{(A_{1636}/A_{1720})_{\text{peak area before curing}}} \right] \times 100\%$$

**2.8 Microtensile bond strength test (μ -TBS)**

The extracted teeth were stored in 1% chloramine T solution, placed at 4 °C, and used
within one month. The tooth was cut perpendicular to the long axis with slow-speed
saw under water cooling in the middle section of the tooth to expose the dentin surface.
Then, the dentin was sanded with 600-grit SiC paper to produce a uniform smear layer
and was ultrasonically cleaned for 5 min. The prepared teeth were randomly divided
into four groups (PU4 adhesive, SB2, SPB, and SBU groups). The specimen was etched
with 37% phosphoric acid gel for 15 s, rinsed for 30 s, and air-blown for 5 s. The
adhesive was applied to the dentin surface using a microbrush, air-thinned for 5 s, and
light-cured for 10 s. **Three 1.5-mm-thick layers of commercial composite resin Z350XT**
**were placed over the surface of the treated dentin. Each resin composite was light cured**
**for 40 s using a light-curing unit.** The specimens were soaked in deionized water at
37 °C for 24 h. After immersion, the specimens were longitudinally cut into sticks of
approximately 1.0 mm in width using a slow speed saw. The dentin-resin stick was
fixed to a microtensile mold using isocyanate glue. Then, it was carried out on a

[revised manuscript text omitted]

PU4, aging (I/J) PU4, immediate. PU4 adhesive can penetrate into the dentinal tubules,
some of which even reach 50 μm . *Dentin(d), Composite resin(c), Adhesive layer(a).

**3.7 Microleakage in composite restoration**

The microleakage of the PU4 adhesive is significantly smaller than that of the other
three commercial adhesives, and the microleakage exceeds half of the depth of the hole
but does not involve the axial surface and only reaches grade 2. However, the
microleakage of three commercial adhesives has been found to involve the axial surface
or even the pulp, reaching grades 3 and 4, as shown in Fig. 8. Thermocycling is often
used in dental adhesive experiments to simulate temperature changes in the oral cavity.
Due to the difference in thermal expansion coefficient between the composite resin and
the tooth, stress will be generated after repeated temperature changes, which eventually
accelerates the formation of microleakage(43). In this study, PU4 adhesive can
significantly reduce the occurrence of microleakage, mainly due to its elastic properties,
which change the rigid connection to elastic bonding between composite resin and
dentin. The stress generated by the inconsistent thermal expansion coefficient can be
buffered through deformation of the PU4 adhesive, reducing the probability of fracture
and secondary caries and improving the stability and durability of the adhesive. The
microleakage of the small molecule dye is qualitative, so more evaluation is needed in
the future study.

Fig. 8 Microleakage between dentin and composite resins after 5000 thermocycling. A
(SB2); B (SPB); C (SBU); D (PU4). Bar = 500 μm .

9 **3.8 Biocompatibility of PU adhesive**

Cytotoxicity was evaluated by examining the effect of PU4 adhesive on cell activity
and morphological changes in L929 fibroblasts. As shown in Fig. 9 (A), there was no
significant difference between the experimental groups and the blank control group in
cell proliferation activity, regardless of the time (24 h, 48 h, 72 h) of extraction. Cell
viability greater than 75% is considered to be noncytotoxic(21). In this experiment, the
cell viability was all greater than 80%.

L929 cells were cultured for 24 h using the 24 h extraction, and their cell viability
was quantified by a live/dead staining experiment. As shown in Fig. 9 (B), the
proportion of living cells was calculated using ImageJ software, and the result is
consistent with that of the MTT experiment. The morphology of L929 cells was directly
observed with an inverted microscope after culture with the 24 h extraction, as shown
in Fig. 9 (B). L929 cells were lengthened and became spindle-shaped, similar to the
morphology of typical fibroblasts in the control group. The cell culture was evenly
distributed, and the intercellular spaces did not change significantly. This indicates that
PU4 adhesive can meet the clinical biosafety demands.

Fig. 9 (A) Cytotoxicity of the PU4 adhesive and commercial adhesives SB2, SPB, and
SBU was determined by MTT assay. A cell relative survival rate greater than 70% can
be considered non-cytotoxic. (B) Optical microscopy images (left) and live/dead
fluorescence cell staining (right) after L929 cells were cultured for 24 h in culture
medium (control), extracted liquid of the PU4 adhesive and commercial dental
adhesives SB2, SPB, SBU. Living cells were stained with calcium—AM (green), and
dead cells were stained with PI (red). Scale bars=100 μm

**4 Conclusion**

In summary, an elastic PU adhesive was prepared and evaluated using comprehensive
methods. The lower water sorption/solubility and decreased microleakage of PU
adhesive indicates that it can prevent water from permeating into the bonding interface
and enhance marginal sealing. The PU adhesive can also buffer the stress coming from
volumetric polymerization shrinkage, temperature changes and repeated chewing force
by deformation due to its elastic property. Furthermore, it is biosafe for L929 cells. This
study lays the foundation for the application of PU adhesive in clinical practice to
produce stable and long-lasting adhesion.

**Conflicts of interest**

There are no conflicts to declare.

**Funding**

This research was financially supported by the National Natural Science Foundation
of China (NSFC, Grant No. 81671033).

**Reference**

- 1. Puckett AD, Fitchie JG, Kirk PC, Gamblin J. Direct composite restorative
materials. *Dent Clin North Am.* 2007;51(3):659-75.
- 2. Khoroushi M, Ehteshami A. Marginal microleakage of cervical composite resin
restorations bonded using etch-and-rinse and self-etch adhesives: two dimensional vs.
three dimensional methods. *Restor Dent Endod.* 2016;41(2):83-90.

3. Correa M, Peres M, Peres K, Horta B, Barros A, Demarco F. Amalgam or composite resin? Factors influencing the choice of restorative material. *J Dent*. 2012;40(9):703-10.
4. Ferracane JL. Resin composite—state of the art. *Dent Mater*. 2011;27(1):29-38.
5. Pashley DH, Tay FR, Breschi L, Tjäderhane L, Carvalho RM, Carrilho M, Tezvergil-Mutluay A. State of the art etch-and-rinse adhesives. *Dent Mater*. 2011;27(1):1-16.
6. Van Meerbeek B, Yoshihara K, Yoshida Y, Mine A, De Munck J, Van Landuyt KL. State of the art of self-etch adhesives. *Dent Mater*. 2011;27(1):17-28.
7. Breschi L, Mazzoni A, Ruggeri A, Cadenaro M, Di Lenarda R, Dorigo E. Dental adhesion review: aging and stability of the bonded interface. *Dent Mater*. 2008;24(1):90-101.
8. Spencer P, Ye Q, Park J, Topp EM, Misra A, Marangos O, Wang Y, Bohaty BS, Singh V, Sene F, Eslick J, Camarda K, Katz JL. Adhesive/dentin interface: the weak link in the composite restoration. *Ann Biomed Eng*. 2010;38(6):1989-2003.
9. Duarte Jr S, Dinelli W, Silva MH. Influence of resin composite insertion technique in preparations with a high C-factor. *Quintessence Int*. 2007;38(10):829-35.
10. Weinmann W, Thalacker C, Guggenberger R. Siloranes in dental composites. *Dent Mater*. 2005;21(1):68-74.
11. Boaro LC, Gonçalves F, Guimarães TC, Ferracane JL, Versluis A, Braga RR. Polymerization stress, shrinkage and elastic modulus of current low-shrinkage restorative composites. *Dent Mater*. 2010;26(12):1144-50.
12. Alnazzawi A, Watts DC. Simultaneous determination of polymerization shrinkage, exotherm and thermal expansion coefficient for dental resin-composites. *Dent Mater*. 2012;28(12):1240-9.
13. Yingchao Z, Haihuan G, Dan F, Tengjiaozi F, Danfeng C, Zuosen S, Song Z, Zhanchen C. New strategy for overcoming microleakage: an elastic layer for dental caries restoration. *J Mater Chem B*. 2015;3(21):4401-5.
14. Ferracane JL. Models of caries formation around dental composite restorations. *J Dent Res*. 2017;96(4):364-71.
15. Van Landuyt KL, Snauwaert J, De Munck J, Peumans M, Yoshida Y, Poitevin A, Coutinho E, Suzuki K, Lambrechts P, Van Meerbeek B. Systematic review of the

chemical composition of contemporary dental adhesives.
Biomaterials. 2007;28(26):3757-85.
16. Nishitani Y, Yoshiyama M, Donnelly A, Agee K, Sword J, Tay F, Pashley DH.
Effects of resin hydrophilicity on dentin bond strength. J Dent Res. 2006;85(11):1016-
21.
17. Tjäderhane L. Dentin bonding: can we make it last? Oper Dent. 2015;40(1):4-18.
18. Ito S, Hashimoto M, Wadgaonkar B, Svizero N, Carvalho RM, Yiu C, Rueggeberg
FA, Foulger S, Saito T, Nishitani Y, Yoshiyama M, Tay FR, Pashley DH. Effects of
resin hydrophilicity on water sorption and changes in modulus of elasticity.
Biomaterials. 2005;26(33):6449-59.
19. Malacarne J, Carvalho RM, Mario F, Svizero N, Pashley DH, Tay FR, Yiu
CK, Carrilho MR. Water sorption/solubility of dental adhesive resins. Dent Mater.
2006;22(10):973-80.
20. Argolo S, Mathias P, Aguiar T, Lima A, Santos S, Foxton R, Cavalcanti A. Effect
of agitation and storage temperature on water sorption and solubility of adhesive
systems. Dent Mater J. 2015;34(1):1-6.
21. Cao D, Zhang Y, Li Y, Shi X, Gong H, Feng D, Guo X, Shi Z, Zhu S, Cui Z.
Fabrication of superhydrophobic coating for preventing microleakage in a dental
composite restoration. Mater Sci Eng C Mater Biol Appl. 2017;78:333-40.
22. Gong H, Guo X, Cao D, Gao P, Feng D, Zhang X, Shi Z, Zhang Y, Zhu S, Cui Z.
Photopolymerizable and moisture-curable polyurethanes for dental adhesive
applications to increase restoration durability. J Mater Chem B. 2019;7(5):744-54.
23. Ferracane JL. Hygroscopic and hydrolytic effects in polymer networks. Dent Mater.
2006;22(3):211-22.
24. Santerre J, Shajii L, Leung BW. Relation of dental composite formulations to their
degradation and the release of hydrolyzed polymeric-resin-derived products. Crit Rev
Oral Biol Med. 2001;12(2):136-51.
25. Soles CL, Yee AF. A discussion of the molecular mechanisms of moisture
transport in epoxy resins. J Polym Sci B: Polym Phys. 2000;38(5):792-802.
26. Soles CL, Chang FT, Bolan BA, Hristov HA, Gidley DW, Yee AF. Contributions
of the nanovoid structure to the moisture absorption properties of epoxy resins. J Polym
Sci B: Polym Phys. 1998;36(17):3035-48.

- 27. Vanlandingham M, Eduljee R, Gillespie Jr J. Moisture diffusion in epoxy systems.
- J Appl Polym Sci. 1999;71(5):787-98.
- 28. Adamson MJ. Thermal expansion and swelling of cured epoxy resin used in
- graphite/epoxy composite materials. J Mater Sci. 1980;15(7):1736-45.
- 29. Ping Z, Nguyen Q, Chen S, Zhou J, Ding YD. States of water in different
- hydrophilic polymers—DSC and FTIR studies. Polymers. 2001;42(20):8461-7.
- 30. Liu M, Wu P, Ding Y, Li S. Study on diffusion behavior of water in epoxy resins
- cured by active ester. Phys Chem Chem Phys. 2003;5(9):1848-52.
- 31. Sideridou I, Tserki V, Papanastasiou G. Study of water sorption, solubility and
- modulus of elasticity of light-cured dimethacrylate-based dental resins. Biomaterials.
- 2003;24(4):655-65.
- 32. Ferracane J, Berge H, Condon J. In vitro aging of dental composites in water-effect
- of degree of conversion, filler volume, and filler/matrix coupling. J Biomed Mater Res.
- 1998;42(3):465-72.
- 33. Brazel CS, Peppas NA. Mechanisms of solute and drug transport in relaxing,
- swellable, hydrophilic glassy polymers. Polymers. 1999;40(12):3383-98.
- 34. Brazel CS, Peppas NA. Dimensionless analysis of swelling of hydrophilic glassy
- polymers with subsequent drug release from relaxing structures. Biomaterials.
- 1999;20(8):721-32.
- 35. Bouillaguet S, Wataha JC, Hanks CT, Ciucchi B, Holz J. In vitro cytotoxicity and
- dentin permeability of HEMA. J Endod. 1996;22(5):244-8.
- 36. Gerzina T, Hume W. Diffusion of monomers from bonding resin-resin composite
- combinations through dentine in vitro. J Dent. 1996;24(1-2):125-8.
- 37. Xu W, Zhang R, Liu W, Zhu J, Dong X, Guo H, Hu G. A multiscale investigation
- on the mechanism of shape recovery for IPDI to PPDI hard segment substitution in
- polyurethane. Macromolecules. 2016;49(16):5931-44.
- 38. Akindoyo JO, Beg M, Ghazali S, Islam M, Jeyaratnam N, Yuvaraj AR.
- Polyurethane types, synthesis and applications-a review. RSC Adv.
- 2016;6(115):114453-82.
- 39. Burgess J, Cakir D. Comparative properties of low-shrinkage composite resins.
- Compend Contin Educ Dent. 2010;31:10-5.

40. Braga RR, Koplín C, Yamamoto T, Tyler K, Ferracane JL, Swain MV. Composite
polymerization stress as a function of specimen configuration assessed by crack
analysis and finite element analysis. *Dent Mater.* 2013;29(10):1026-33.
41. Carrera CA, Lan C, Escobar-Sanabria D, Li Y, Rudney J, Aparicio C, Fok A. The
use of micro-CT with image segmentation to quantify leakage in dental restorations.
*Dent Mater.* 2015;31(4):382-90.
42. Hong L, Wang Y, Wang L, Zhang H, Na H, Zhang Z. Synthesis and
characterization of a novel resin monomer with low viscosity. *J Dent.* 2017;59:11-7.
43. Wahab FK, Shaini FJ, Morgano SM. The effect of thermocycling on microleakage
of several commercially available composite Class V restorations in vitro. *J Prosthet*
*Dent.* 2003;90(2):168-74.

Authors' Contributions

The authors meet all of the following criteria:

- 1) substantial contributions to conception and design, or acquisition of data, or analysis and interpretation of data;
- 2) drafting the article or revising it critically for important intellectual content;
- 3) final approval of the version to be published; and
- 4) agreement to be accountable for all aspects of the work in ensuring that questions related to the accuracy or integrity of any part of the work are appropriately investigated and resolved.

Authors' major contribution:

Jiahui Zhang: conducted the statistical analysis and wrote this manuscript;

Xiaowei Guo and Xiaomeng Zhang: finish the acquisition of data;

Huimin Wang, Jiufu Zhu and Zuosen Shi: conduct the experiments and repeated them;

Corresponding author: Song Zhu and Zhanchen Cui: designed and oversaw the study, and reviewed and revised this manuscript.

PU elastic dental adhesive

185x107mm (300 x 300 DPI)

Appendix C

Response to the reviewers' comments

Dear Editor and Reviewers,

We hope you are keeping well at this difficult time.

Thanks very much for taking your time to review this manuscript. I really appreciate all these precious comments and suggestions. Please find my itemized responses in below and my revisions in the resubmitted files.

Thanks again.

Reviewer comments to Author:

Reviewer: 2

Comments to the Author(s)

Many of the responses to the reviewer are information that the reader needs to understand the manuscript, especially the parts marked with yellow (see attach file). Authors have to include and/or adapt these parts in the paragraphs of their main text for better clarity of the paper.

We are grateful for the suggestion.

We have adapted relevant parts in the paragraphs of the main text according to the suggestion for better clarity of the paper. The revised section of the manuscript was marked in red.